# Dynamic organization of cerebellar climbing fiber response and synchrony in multiple functional components reduces dimensions for reinforcement learning

**Huu Hoang[1†], Shinichiro Tsutsumi[2†], Masanori Matsuzaki[3], Masanobu Kano[4,5], Mitsuo Kawato[6], Kazuo Kitamura[7]\*, Keisuke Toyama[1]\***

[1]ATR Neural Information Analysis Laboratories, Kyoto, Japan; [2]RIKEN Center for Brain Science, Wako, Japan; [3]Department of Physiology, The University of Tokyo, Tokyo, Japan; [4]Department of Neurophysiology, The University of Tokyo, Tokyo, Japan; [5]International Research Center for Neurointelligence (WPI-IRCN), The University of Tokyo, Tokyo, Japan; [6]ATR Brain Information Communication Research Laboratory Group, Kyoto, Japan; [7]Department of Neurophysiology, University of Yamanashi, Kofu, Japan

**\*For correspondence:**
kitamurak@yamanashi.ac.jp (KK);
toyama@atr.jp (KT)

[†]These authors contributed equally to this work

**Competing interest:** The authors declare that no competing interests exist.

**Abstract** Cerebellar climbing fibers convey diverse signals, but how they are organized in the compartmental structure of the cerebellar cortex during learning remains largely unclear. We analyzed a large amount of coordinate-localized two-photon imaging data from cerebellar Crus II in mice undergoing 'Go/No-go' reinforcement learning. Tensor component analysis revealed that a majority of climbing fiber inputs to Purkinje cells were reduced to only four functional components, corresponding to accurate timing control of motor initiation related to a Go cue, cognitive error-based learning, reward processing, and inhibition of erroneous behaviors after a No-go cue. Changes in neural activities during learning of the first two components were correlated with corresponding changes in timing control and error learning across animals, indirectly suggesting causal relationships. Spatial distribution of these components coincided well with boundaries of Aldolase-C/zebrin II expression in Purkinje cells, whereas several components are mixed in single neurons. Synchronization within individual components was bidirectionally regulated according to specific task contexts and learning stages. These findings suggest that, in close collaborations with other brain regions including the inferior olive nucleus, the cerebellum, based on anatomical compartments, reduces dimensions of the learning space by dynamically organizing multiple functional components, a feature that may inspire new-generation AI designs.

## Editor's evaluation

This is an important study of changes in the amplitude and synchrony of calcium responses in Purkinje cells over the course of learning, measured across a large region of the cerebellar cortex. The evidence for the conclusions is compelling, supported by sophisticated data analysis approaches. This work has the potential to inform our understanding of the functional organization of the cerebellum and longstanding hypotheses about the role of cerebellar climbing fibers in the induction of learning and in the timing of movement.

## Introduction

Computational learning theory asserts that machine learning algorithms necessitate as many training data samples as the number of their parameters for correct generalization (*Watanabe, 2009*; *LeCun et al., 2015*; *Mnih et al., 2015*; *Silver et al., 2016*; *Wurman et al., 2022*). In contrast, although the cerebellum contains tens of billions of neurons and even more synapses, and is involved in diverse functions (*Strick et al., 2009*; *Sokolov et al., 2017*), each of which may require a different coding scheme (*De Zeeuw and Ten Brinke, 2015*; *De Zeeuw, 2021*; *De Zeeuw et al., 2021*), animals can learn new behaviors within thousands of trials, for which the cerebellum is mainly responsible (*Adolph et al., 2012*). To reconcile these observations, previous theories have proposed that cerebellar compartments (*Andersson and Oscarsson, 1978*; *Ito, 1997*; *Nisimaru et al., 2013*) and spike synchronization drastically reduce effective numbers of learning parameters (degrees of freedom) and enable learning from small samples (*Kawato et al., 2011*; *Tokuda et al., 2017*; *Cortese et al., 2019*; *Kawato et al., 2021*). On one hand, the cerebellar cortex is organized into multiple longitudinal compartments with different intrinsic neuronal activity (*Andersson and Oscarsson, 1978*; *Ito, 2013*; *Apps and Garwicz, 2005*; *Apps et al., 2018*; *Sugihara and Shinoda, 2004*; *Sugihara et al., 2007*; *Voogd and Ruigrok, 2004*; *Kano et al., 1990*), each forming a computational unit (*Zhou et al., 2014*). Outstandingly in each of these, aldolase C (AldC)-positive and negative zones represent distinct information in adaptation of the vestibulo-ocular reflex (*De Zeeuw and Koekkoek, 1997*; *Wylie et al., 1994*; *Schonewille et al., 2006*) and in eyeblink conditioning (*Hesslow, 1994a*; *Hesslow, 1994b*; *Mostofi et al., 2010*). Therefore, this compartmentalization constitutes, at least in part, dimension reduction for cerebellar functions related to eye movement and conditioning. On the other hand, synchronization of climbing fiber (CF) activity, which mainly originates from inferior olivary neurons (*Llinas et al., 1974*; *Lang et al., 1999*; *Blenkinsop and Lang, 2006*), contributes strongly to cerebellar functions (*Eccles et al., 1966*; *Nietz et al., 2017*; *Sedaghat-Nejad et al., 2022*). During acquisition of complex behaviors in motor activity (*Welsh et al., 1995*; *Hoogland et al., 2015*; *Wagner et al., 2021*) or reward processing (*Wagner et al., 2017*; *Heffley and Hull, 2019*; *Heffley et al., 2018*; *Kostadinov et al., 2019*), CF synchrony might provide flexible dimension control of neural dynamics (*Kawato et al., 2011*; *Tokuda et al., 2017*; *Kawato et al., 2021*; *Hoang et al., 2020a*). Moreover, synchrony is high within cerebellar compartments, but not across them, implying a structure-function relationship between cerebellar compartments and synchronization (*Ozden et al., 2009*; *De Gruijl et al., 2014*; *Tsutsumi et al., 2015*). However, it is unclear how compartments and synchronization together facilitate dimension reduction for learning of cognitive functions, for which genetic prewiring is unlikely to be sufficient.

To shed light on mechanisms of dimension reduction in the cerebellum, we systematically examined two-photon recordings of CF inputs to Purkinje cells in eight AldC compartments of mice learning an auditory discrimination Go/No-go task (*Tsutsumi et al., 2019*). This task requires reinforcement learning guided by reward, and necessitates, 1. accurate timing control of licking to obtain a reward, 2. learning based on reward prediction error, 3. reward processing, and 4. inhibition of licking to the No-go-cue. We applied a hyper-acuity spike timing detection algorithm with 10 ms resolution (*Hoang et al., 2020b*) to estimate timing of complex spikes (CSs), which were reliably recorded by calcium imaging. Tensor component analysis (*Williams et al., 2018*) of CS activity revealed that 50% of the variance was explained by only four clearly separable components (TC1-4), each corresponding to one of the above four functions. Furthermore, spatial distributions of these four components were markedly different, with functional boundaries between the medial and lateral Crus II and across AldC compartments. Ten-millisecond (ms) resolution analyses revealed that CS synchrony of a specific component was high for a specific engaged cue-response condition, and was spatially localized within corresponding compartments. Tensor components were shaped by bidirectional synchrony-response changes during the course of learning. These results demonstrate that the cerebellum reduces dimensions of activity in a large number of neurons to a much smaller number of components by a synchronization scheme that conforms to a specific task. Behaviorally, across animals, changes in CS synchrony during learning of TC1 neurons were correlated with changes in lick fluctuation in Go trials. In regard to learning, changes in relative proportion of TC2 neurons were correlated with changes in performance of No-go trials. These results indirectly suggested causal relationships of TC1 with timing control and TC2 with cognitive error learning. Interestingly, we also found that individual anatomical compartments and even single PCs can contain signals from multiple components (*Kitazawa et al., 1998*; *Sendhilnathan et al., 2021*; *Ikezoe et al., 2023*; *Markanday et al., 2021*). This study therefore

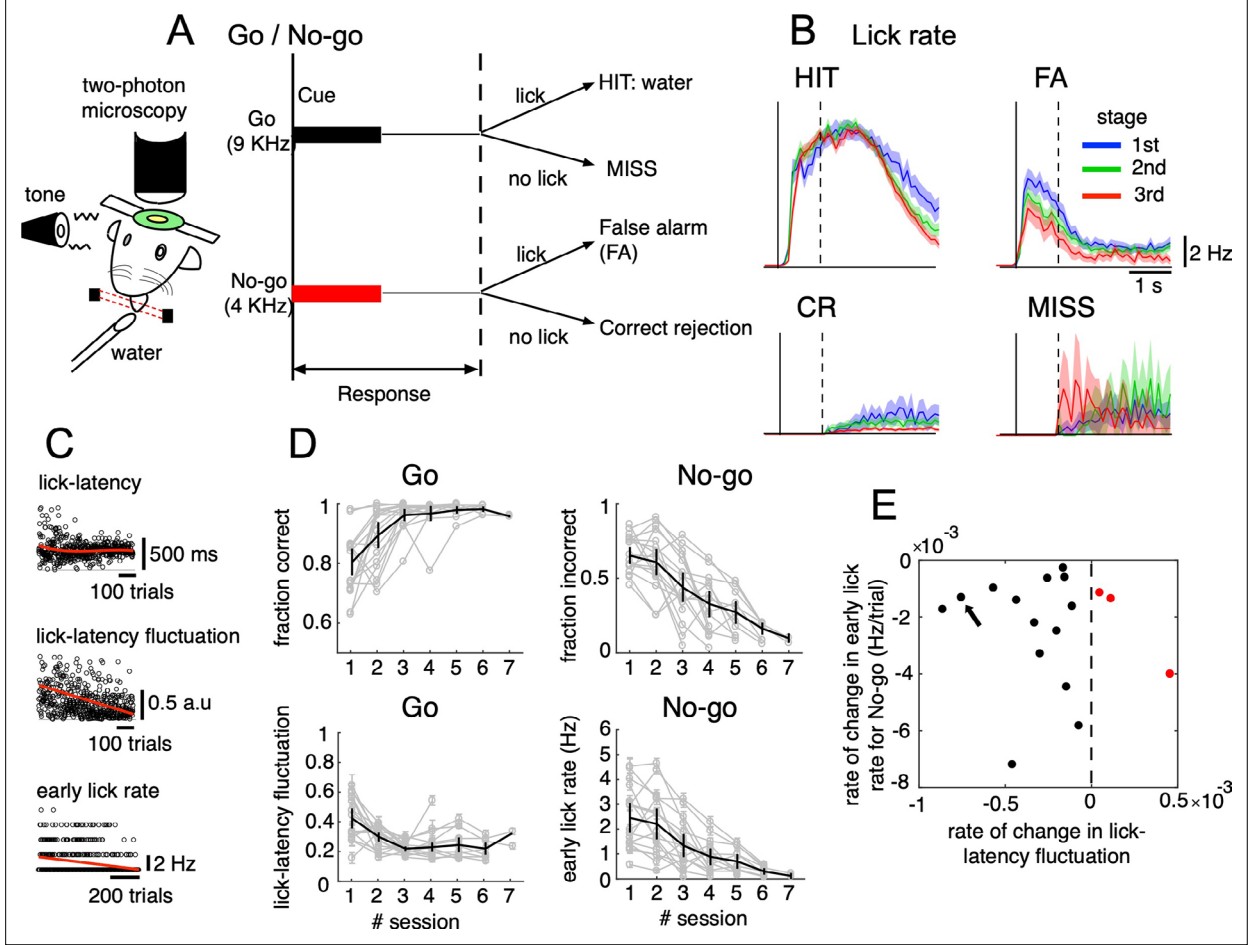

**Figure 1.** Go/No-go auditory-cue discrimination task and behavior changes during learning. (**A**) Schematic diagram of a mouse performing the Go/No-go discrimination task under a two-photon microscope. (**B**) the lick rate of the four cue-response conditions sampled in the three learning stages (blue, green and red traces for 1st, 2nd, and 3rd stages, respectively). Thick lines and shadings represent mean ± s.e.m (n=12,334, 5,588, 7,681 and 914 trials for HIT, FA, CR and MISS, respectively). Solid and dashed vertical lines in **A**-**B** indicate the timing of cue onset and end of the response window, respectively. (**C**) from top to bottom, lick-latency, lick-latency fluctuation in Go trials and the early lick rate in No-go trials of a representative mouse (indicated by black arrow in **E**). Trials were sorted by the time course of training. Red traces indicate polynomial fittings of lick parameters as functions of trials (see Methods). (**D**) changes in four learning indices as functions of training sessions, including the fraction correct of Go cues, the fraction incorrect of No-go cues, lick-latency fluctuation in Go trials, and the early lick rate in No-go trials. Thin gray traces represent individual animals (n=17) and thick dark traces with error bars represent mean ± s.e.m across all animals. (**E**) scatterplot for rate of change in lick-latency fluctuation for Go cues (abscissa) and rate of change in early lick rate for No-go cues (ordinate) estimated from licking behavior of individual mice. Black dots were for mice whose rates were both negative and red dots were for the three mice that showed increased lick-latency fluctuation after learning (positive rate).

The online version of this article includes the following source data and figure supplement(s) for figure 1:

**Source data 1.** Datasets used to create *Figure 1*.

**Figure supplement 1.** Licking behavior in the early response window for individual mice.

provides the first evidence that distinct neural mechanisms, which are related to dynamic changes in climbing fiber responses and CS synchrony, can function simultaneously in multiple cerebellar components, thus serving a variety of cerebellar functions in learning of a single task (*Kawato et al., 2021*; *Llinás, 2013*; *Lang et al., 2017*; *Marr, 1969*; *Albus, 1971*; *Ito, 1970*; *Ito, 2002*).

## Results

### Two types of behavioral learning in a Go/No-go task

We trained mice to perform a Go/No-go auditory discrimination task (*Figure 1A*) that required not only sensorimotor-related functions, but also cognitive functions, such as response inhibition (*Tsutsumi*

et al., 2019). Briefly, mice (n=17) were trained to associate the 'Go' cue (a 9 kHz tone for 0.5 s) with a licking during a response period of 1 s after cue onset to get a water reward. The 'No-go' cue (a 4 kHz tone for 0.5 s) was not associated with a reward, but a 4.5 s timeout was imposed if the mice licked during the response period (Figure 1A). We recorded 87 sessions from 17 mice (each underwent no more than 7 sessions), including 26,517 trials of auditory discrimination Go/No-go tasks in the three learning stages (1st, 2nd, and 3rd stages with fraction correct <0.6, 0.6–0.8, 0.8<, respectively). We categorized trials into the four cue-response conditions: HIT trials in the Go task (lick after Go cue, n=12,334), false alarm (FA) trials (lick after No-go cue, n=5,588), correct rejection (CR) trials (no lick after No-go cue, n=7,681) and MISS trials (no lick after Go cue, n=914). Note that before the main Go/No-go auditory discrimination training, mice had already been trained to lick in response to both high and low tones with indiscriminate reward feedback.

Behavioral data indicated that the lick rate was dramatically reduced in FA trials during the three stages of learning (Figure 1B), but it remained relatively unchanged in HIT trials. During learning, there was an increase in the fraction correct for the Go cue (0.78±0.1 and 0.98±0.01, 1st and 6th sessions, respectively, Figure 1D). There were no consistent changes in the reaction time after Go; however, we found decreased lick-latency fluctuation in the licking response with learning (0.45±0.15 and 0.22±0.06, 1st and 6th sessions, respectively, Figure 1C–D). For the No-go cue, the fraction incorrect decreased after learning (0.66±0.12 and 0.15±0.06, 1st and 6th sessions, respectively, Figure 1D), along with a decreased lick rate in the early response window (0–500ms after cue onset, 2.4±1.2 and 0.3±0.2 Hz, 1st and 6th sessions, respectively, Figure 1C–D). At an individual level (see Figure 1— figure supplement 1 for behavior changes associated with Go and No-go cues of individual mice), 14 out of 17 mice showed a negative rate of change in lick-latency fluctuation, while the rate of learning-related changes in the number of early licks for the No-go cue was negative for all mice (Figure 1E). In summary, behavioral data indicated that mice successfully learned an auditory discrimination task by changing licking behavior in the early response window, first, achieving more precise timing of the first lick after Go cues and second, suppressing licks after No-go cues.

## Opposite changes in cue-related CF responses in medial and lateral parts of Crus II during learning

We analyzed two-photon recordings of CF-dependent dendritic Ca²⁺ signals, a later-learning stage, part of which was reported previously (Tsutsumi et al., 2019), focusing on how they change within and across AldC compartments as learning proceeds. Briefly, we used the Aldoc-tdTomato transgenic mouse line (Tsutsumi et al., 2015) to systematically explore functional differences in CF inputs to AldC compartments at single-cell resolution during the task. While mice underwent the Go/No-go task, we performed 236 sessions of two-photon calcium imaging (sampling rate, 7.8 Hz) from PC dendrites at every boundary of AldC expression in eight AldC compartments (7+, 6-, 6+, 5-, 5+, 5a-, 5a+and 4b-) in lobule Crus II to simultaneously monitor calcium responses (see Methods and Supplementary file 1 for detailed numbers of neurons and trials recorded). To investigate calcium responses with higher temporal resolution than the two-photon recordings, timing of complex spikes (CSs) was estimated for 6,445 PCs using hyperacuity software (Hoang et al., 2020b) (HA_time, 100 Hz, see Methods and Figure 2—figure supplement 1 for reliability of HA_time). These technical advances allowed us to monitor CS activities of a large number of Purkinje cells in different AldC compartments during the course of learning with high temporal precision.

We studied population peri-stimulus time histograms (PSTHs) of CSs sampled in the three learning stages for the four cue-response conditions. Note that spontaneous CS activities (around 1 Hz) were subtracted from the PSTHs (see Methods). CS activity in HIT trials was relatively strong in the 1st stage (peak PSTH, 1.7 Hz) and slightly increased in the 3rd stage (peak PSTH, 2.3 Hz, Figure 2A). In contrast, CS activity in FA trials was strong initially, but it was significantly reduced after learning (peak PSTH, 2.5 and 0.9 Hz for 1st and 3rd stage, respectively, Figure 2B). CS activity in CR trials was almost unchanged during learning (peak PSTH, 0.7 and 0.9 Hz for 1st and 3rd stage, respectively, Figure 2C). For MISS trials, only spontaneous CSs were observed (Figure 2D). While investigating PSTHs for individual AldC compartments, we found contrastive response profiles between the lateral (AldC 7+to 5-) and medial parts of Crus II (AldC 5+to 4b-) segregated by the anatomical and functional border (Tsutsumi et al., 2019; Figure 2A–D). CS firings in HIT trials were large and diffuse at the initial learning stage across the entire medial Crus II, as well as a fraction (AldC 6+) of the lateral Crus II (Figure 2A,

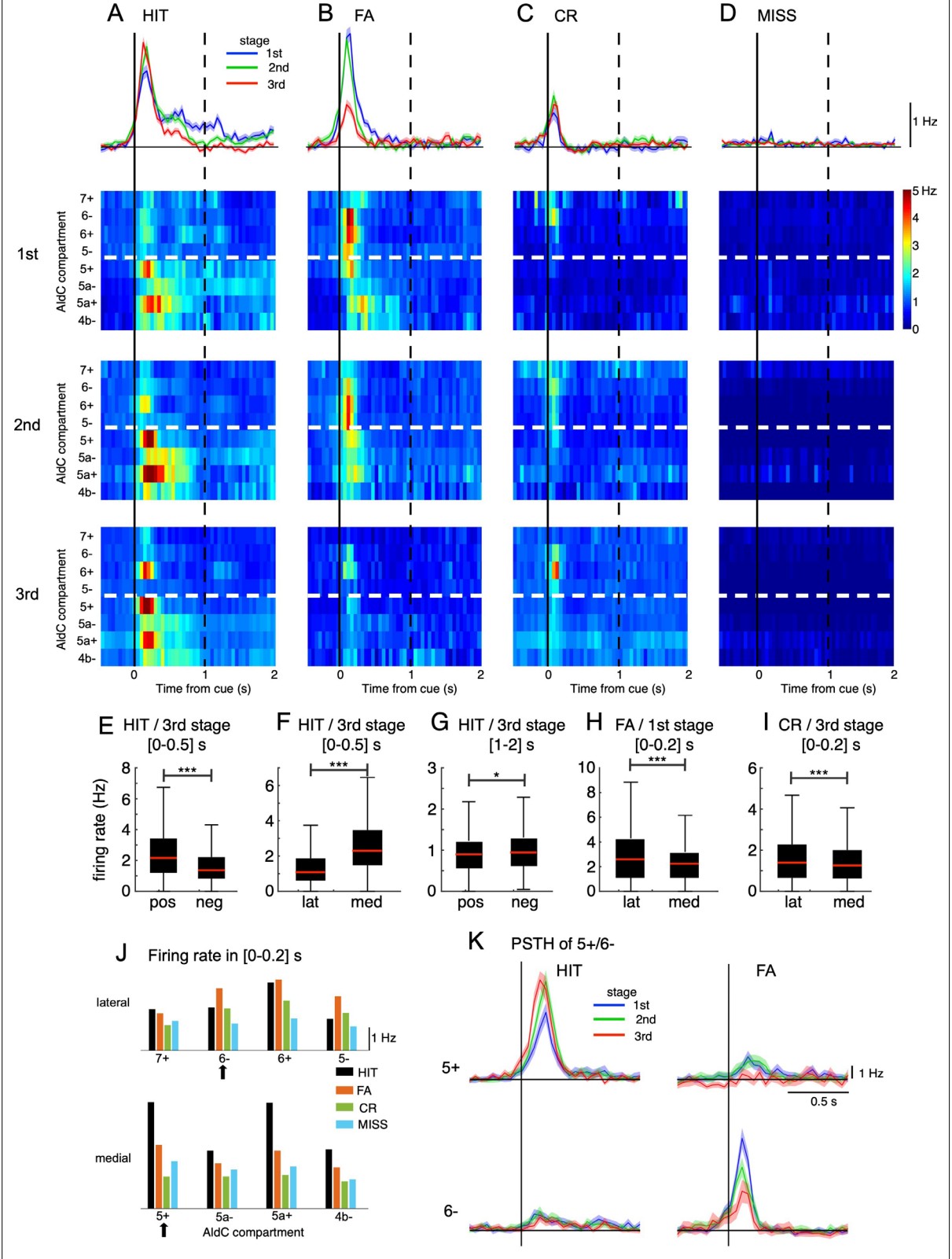

**Figure 2.** Opposite changes in CS firings in the lateral and medial parts of Crus II. (**A–D**) Top row: population peri-stimulus time histograms (PSTHs) of CSs sampled for all recorded neurons (n=6,445) during three learning stages across four cue-response conditions: HIT trials (n=12,334, **A**), FA trials (n=5,588, **B**), CR trials (n=7,681, **C**), and MISS trials (n=914, **D**). From second to bottom rows: pseudo-color representation of PSTHs in each AldC compartment in Crus II sampled in three learning stages. Vertical black solid lines and black dashed lines represent the cue onset and the end of

*Figure 2 continued on next page*

*Figure 2 continued*

response window, respectively. Thick white dashed lines represent the boundary between lateral vs. medial hemispheres of Crus II. (**E–I**) Box plots show CS firing rate estimated for the AldC positive vs. negative compartments and lateral vs. medial Crus II in various configurations of the cue-response condition, learning stages and temporal windows. For each box plot, the black bar indicates the 25% and 75% and the central red mark indicates the median. Asterisks indicate the significance level of one-way ANOVA: * p<0.05, **** p<0.0001. (**J**) CS firing rate sampled in 0–0.2 s after cue onset for HIT (black bars), FA (orange bars), CR (green bars), and MISS (cyan bars) trials across all three learning stages for individual AldC compartments. Black arrows indicate the two representative AldC compartments that show the largest changes in CS activity after learning for HIT (5+) and FA (6-) trials. (**K**) PSTHs in HIT and FA trials of the two representative AldC compartments 5+ (n=1,117 and 575 for HIT and FA trials, respectively) and 6- (n=822 and 408 for HIT and FA trials, respectively) during learning. Thick lines and shadings in (**A-D**) and (**K**) represent mean ± s.e.m.

The online version of this article includes the following source data and figure supplement(s) for figure 2:

**Source data 1.** Datasets used to create *Figure 2*.

**Figure supplement 1.** Illustration of CF reconstruction by HA_time and its examination.

**Figure supplement 2.** Synchrony dynamics and associated synchrony-response bidirectional changes in AldC compartments 5+and 6-.

1st stage). These firings became even stronger and more segregated in AldC positive compartments of the lateral and medial parts of Crus II (AldC 6+, 5+, 5a+), and temporally confined within 200ms after cue onset (2nd and 3rd stages), as learning proceeded. PSTHs in FA trials were as strong as those in HIT trials initially, distributed across almost the entire lateral and medial parts of Crus II, but more in lateral parts (*Figure 2B*, 1st stage). They gradually decreased with learning and finally became rather weak (2nd - 3rd stages), while maintaining the initial compartmental distribution profiles. For CR trials, PSTHs were initially localized in the lateral Crus II, and finally became confined to a fraction (AldC 6-, 6+) of the lateral Crus II (*Figure 2C* 1st - 3rd stages). There were only spontaneous CSs in MISS trials across the entire learning stage (*Figure 2D*).

We found remarkable differences in CS activity between AldC positive and negative compartments as well as between lateral and medial parts of Crus II in various configurations of the cue-response condition, learning stages and temporal windows. In particular, at the third stage, CS firing rate in HIT trials sampled in the [0–0.5] s window was higher for AldC-positive (mean ± std, 2.47±1.77 Hz) and medial (2.70±1.75 Hz) compartments than that for AldC negative (1.69±1.32 Hz) and lateral (1.37±1.05 Hz) compartments, respectively (one-way ANOVA, p<0.0001 for both positive vs. negative and lateral vs. medial compartments, *Figure 2E and F*). However, when sampled in the same learning stage and cue-response condition, but with a delayed temporal window of [1 - 2] s, CS firing rate for AldC negative compartments (1.04±0.60 Hz) was higher than positive counterparts (0.98±0.60 Hz, p=0.02, *Figure 2G*). For FA trials and at the first stage, CS firing sampled in [0–0.2] s window was higher for the lateral (2.98±2.32 Hz) part than the medial part (2.35±1.70 Hz, p<0.0001, *Figure 2H*). Similarly, at the third stage, the lateral Crus II (1.72±1.56 Hz) also showed a higher CS firing rate in CR trials than the medial part (1.40±1.08 Hz, p<0.0001, *Figure 2I*).

We also studied the compartmental topology of cue responsiveness, evaluating the CS firing rate sampled within 0–200ms after cue onset for all three learning stages (*Figure 2J*, see Methods). Response strength in HIT trials (black bars) across compartments exhibited three high compartmental peaks – including compartments 5+ (4.6 Hz), 5a+ (4.5 Hz), and 6+ (2.9 Hz) – and 4 smaller valleys – including compartments 7+ (1.8 Hz), 5- (1.4 Hz), 5a- (2.5 Hz), and 4b- (2.5 Hz). In contrast, response strength in FA trials (orange bars) was stronger in the lateral than the medial Crus II in partial accordance with *Figure 2H*. Response strength in CR (green bars) shows a decline from lateral to medial in partial accordance with *Figure 2I*, and MISS trials remained almost flat across both medial and lateral parts of Crus II (cyan bars).

To better demonstrate opposite response changes of medial and lateral parts at each learning stage, we selected the top 100 neurons in AldC compartments 5+and 6-, which showed the largest differences in CS activity after learning for HIT and FA trials, respectively, i.e., the neurons whose response strength was most selective for HIT or FA trials. We observed that AldC 5+neurons exhibited a marked increase in CS firings as well as a phase advance in cue-response along with learning for the HIT condition (peak PSTH, 6.6 and 9.4 Hz for the 1st and 3rd stages, respectively, *Figure 2K*), whereas that of AldC 6- neurons showed a significant decrease for the FA condition along with learning (8.6 and 4.0 Hz for the 1st and 3rd stages, respectively). Importantly, we found remarkably opposite changes in synchrony with learning of neurons in AldC compartments 5+and 6-, which were strongly associated

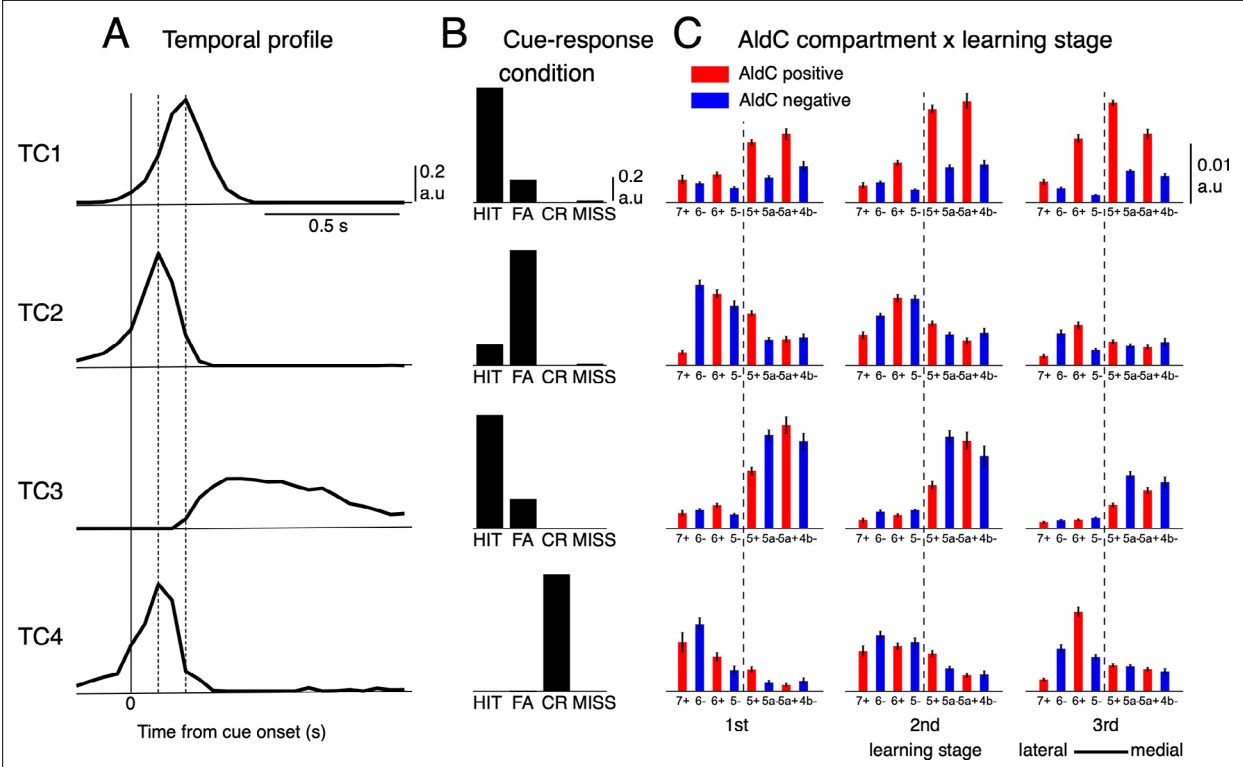

**Figure 3.** Tensor component analysis of CS activity. (**A–C**) (**A**) Coefficients of temporal, (**B**) cue-response condition, and (**C**) neuronal factor of the four tensor components (TC1-TC4, from top to bottom) estimated by TCA. TCs are shown in their contribution order (see Methods for details). The three vertical lines in (**A**) represent the timing at 0, 100 and 200ms after the cue onset. Bars with lines in (**C**) show means and SDs of neuron factor coefficients, grouped by eight AldC compartments (shown in red and blue colors for AldC positive and negative compartments, respectively) and three learning stages (n=6,445 neurons). The thick dashed line in (**C**) indicates a functional boundary between the lateral and medial Crus II (*Tsutsumi et al., 2019*).

The online version of this article includes the following source data and figure supplement(s) for figure 3:

**Source data 1.** Datasets used to create *Figure 3*.

**Figure supplement 1.** Tensor component analysis of population PSTHs with a varied number of tensor components and the permutation test.

with corresponding opposite changes in their responses even on a single-trial basis (*Figure 2—figure supplement 2*). We named them 'bidirectional synchrony-response changes'.

## Tensor component analysis of CF activities

Next, we examined whether bidirectional synchrony-response changes observed in AldC compartments 5+and 6- can be generalized across the entire Crus II. For this purpose, we conducted tensor component analysis (*Williams et al., 2018*) (TCA) to decompose high-dimensional CS firings, that is, PSTHs of neurons in the four cue-response conditions sampled during three different learning stages, into low-dimensional components, each with a unique set of coefficients of the temporal profile, the cue-response condition, and the neuron (see Methods for more details).

We found that only four distinct tensor components (TCs) underlie CS firings (*Figure 3*), and they accounted for about half the variance among 6,445 neural PSTHs (see *Figure 3—figure supplement 1* for more details). The first TC (TC1) had a fast temporal profile peaking 200ms after cue onset (*Figure 3A*). This component, TC1, was dominant mostly in the HIT condition and only weakly manifested in FA (*Figure 3B*). Compartmentally, TC1 was concentrated in AldC positive compartments across the medial and lateral Crus II and gradually increased its coefficients along with learning (*Figure 3C*). In contrast, the second TC (TC2) was dominant in the FA condition and weakly manifested in HIT (*Figure 3A*). It had a very fast temporal profile that peaked 100ms after cue onset, and decayed sharply toward baseline within 400ms after cue onset. TC2 was distributed broadly in the lateral Crus II during early stages of learning (*Figure 3C*), but its coefficients were markedly reduced in the 3rd stage. The third TC (TC3) had a slow temporal profile, peaking at ~300ms, and was prolonged for 1 s

after cue onset. TC3 was dominant in the HIT condition and little observed in FA. Compartmentally, TC3 was initially distributed in the entire medial Crus II, but mainly in AldC negative compartments during later stages of learning (*Figure 3C*). Finally, the fourth TC (TC4) had a temporal profile and compartmental distribution similar to those of TC2, but was present only in the CR condition. In summary, TCA revealed four TCs with distinct temporal and compartmental profiles for different cue-response conditions.

## Synchrony dynamics shape tensor component activities

To investigate synchrony dynamics of the four TCs, we selected neurons that most strongly represented each component, that is, with the highest contribution to that component. Briefly, we sampled the top 300 neurons for each TC at each learning stage, then removed overlaps. As a result, we selected 2096 neurons (termed 'topTC neurons'), whose cumulative responses accounted for approximately 40% of all responses in the entire population (see Methods and *Figure 4—figure supplement 1* for more details; Basically, the same results were obtained when we selected a fixed proportion of sampled neurons for each TC at each learning stage). Firings for each class of topTC neurons varied considerably with learning stages, while maintaining roughly the same temporal profiles of the corresponding TCs (compare *Figure 4—figure supplement 1C* PSTH temporal profiles at the three learning stages with the corresponding temporal profiles of TCs in *Figure 3A*).

We found that synchrony strengths (estimated as the summation of cross-correlograms in ± 10ms around the center time bin, see Methods) within topTC neurons (0.40±0.17, 0.36±0.17, 0.33±0.17 and 0.19±0.13 for TC1-4, respectively) were significantly larger than those across topTCs (0.12±0.09, p<0.0001 for all pairwise comparisons between within- and across- topTC populations, *Figure 4A*). We further investigated synchrony strengths for specific cue-response conditions in which TC1-2 were engaged. More specifically, synchrony strength (which was normalized by the number of spikes) of topTC1 neurons in HIT trials (0.40±0.17) and that of topTC2 neurons in FA trials (0.37±0.17) were significantly stronger than those in other cue-response conditions (0.19±0.11 and 0.23±0.14 for topTC1-others and topTC2-others, respectively, *Figure 4B*). This tendency was observed even on a single-trial basis, and strong instantaneous synchrony (total number of synchronous firings in 30 ms bins in a window of 300ms before the first lick onset, see Methods) was found in TC1-HIT and TC2-FA trials at the and learning stages, respectively (*Figure 4C* and *Video 1*). Furthermore, such synchrony dynamics were well aligned with the spatial distribution of topTC1-2 (see *Video 2*). Together, these results suggested that synchronization, spatially guided by AldC compartments, organizes TC populations only during dedicated cue-response conditions.

Moreover, we found that firing of topTC1 neurons strongly increased (CS firing rate in 0–200ms window, mean ± std, 5.3±1.8, 7.7±2.4 and 8.0±1.9 Hz for 1st, 2nd and 3rd stage; one-way ANOVA, p<0.0001, *Figure 4D*) along with learning for the HIT condition, while it was moderate at the 1st stage and almost disappeared at the 3rd stage for the FA condition. In contrast, the cue-response of topTC2 neurons remained small across all learning stages for the HIT condition, while it was strong at the 1st and 2nd stages and markedly decreased at the 3rd stage for the FA condition (5.4±2.2, 5.1±2.0 and 3.7±2.1 Hz for 1st, 2nd and 3rd stage; one-way ANOVA, p<0.0001, *Figure 4D*). We also observed a significant increase in synchrony strength of HIT trials during learning of topTC1 neurons (mean ± std, 0.33±0.16, 0.45±0.16 and 0.46±0.17 for 1st, 2nd, and 3rd stage; one-way ANOVA, p<0.0001, *Figure 4E*) and a significant decrease in synchrony strength of FA trials of topTC2 neurons (0.43±0.18, 0.37±0.16 and 0.28±0.17 for 1st, 2nd and 3rd stage; one-way ANOVA, p<0.0001). Together, these findings suggest that two opposite response changes in the early window associated with the Go cue for TC1 and the No-go cue for TC2 were accompanied by corresponding synchrony changes (see *Figure 4—figure supplement 2* for CS synchrony of the four TCs).

## Correlations between CF synchrony and licking behaviors

The fact that topTC1 neurons increased synchrony in HIT trials during learning of precisely timed initiation of licking and that topTC2 neurons decreased synchrony in FA trials along with reduction in erroneous licks with learning suggest that two opposite synchrony changes are related to two changes in licking behaviors for the two corresponding cue-response conditions. To test these possibilities, we investigated synchrony-behavior correlations in the three response windows, namely, early lick (0–0.5 s), reward lick (0.5–2 s) and succeeding lick (2–4 s). These windows were determined from

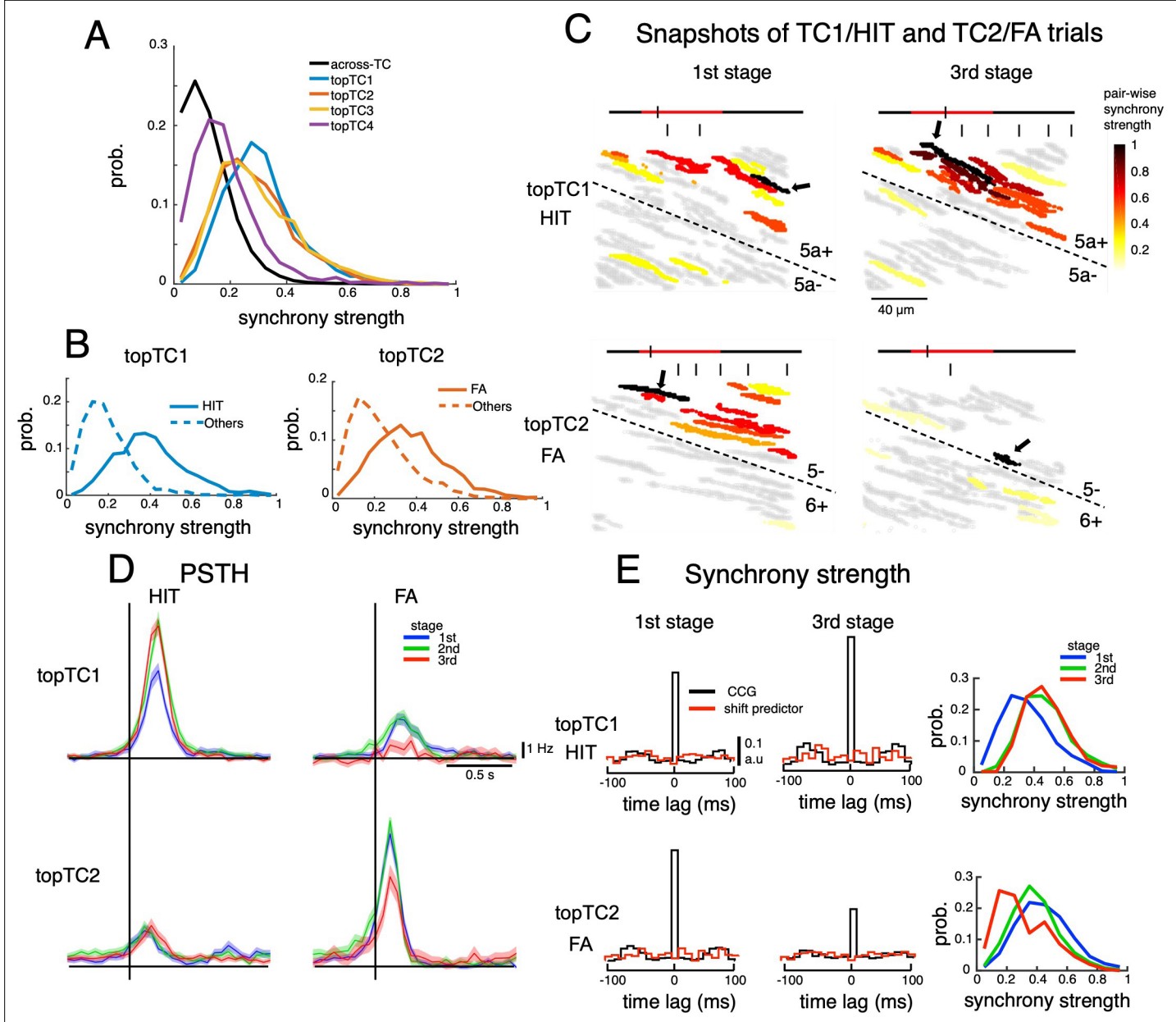

**Figure 4.** Dynamics of synchrony and opposite changes in synchrony of TC1-TC2 neurons during the course of learning. (**A**) Histograms of synchrony strength between topTC1-4 neurons (colored traces, n=508, 461, 581 and 546 for topTC1-4, respectively) in comparison with those across TCs (black trace). (**B**) Histograms of synchrony strength computed specifically in HIT trials for topTC1 neurons (solid blue trace) and in FA trials for topTC2 neurons (solid orange trace), contrasting with those in other cue-response conditions (dashed traces). (**C**) Representative images of synchronous firings in TC1/HIT (5a-/5a+) and TC2/FA (6+/5-) in the 1st and 3rd stages. The horizontal line shows the time course of –200ms to 1 s after cue onset with a small tick indicating the timing of the snapshot relative to the cue onset (the cue period of 500ms was shown in red color). Short vertical bars indicate lick timings. Snapshots capture responses of Purkinje cell dendrites (gray areas) co-activated in a time bin of 10ms. The hot-color spectrum represents pair-wise synchrony strength between reference cells (pointed by black arrows) and other cells in the same recording session (see *Video 1*). (**D**) PSTHs of topTC1-TC2 neurons in HIT (n=1,983 and 1,703 for topTC1 and topTC2, respectively) and FA (n=1,010 and 813 for topTC1 and topTC2, respectively) trials in three learning stages. (**E**) Population cross-correlograms (CCGs) of the 1st and 3rd stages indicated opposite changes in synchrony strength of topTC1-TC2 neurons during learning. Red traces in CCGs indicate shift predictors estimated for the correlation solely due to the cue stimulus. Solid lines in the right panels represent histograms of synchrony strength within topTC1 and topTC2 neurons in HIT and FA trials, respectively, for the three learning stages.

The online version of this article includes the following source data and figure supplement(s) for figure 4:

**Source data 1.** Datasets used to create *Figure 4*.

*Figure 4 continued on next page*

*Figure 4 continued*

**Figure supplement 1.** Neural sampling by TCA.

**Figure supplement 2.** Synchronous firings of TCs.

behavioral data showing that rewards were given in a window of 0.41–1.23 s after the first lick, timing of which was about 0.5 s after cue onset.

For the early lick window, we found a tendency for topTC1 synchronous CS-triggered lick responses in Go trials to be mostly positive, peaking at 200–300ms for all three learning stages (*Figure 5A*). Importantly, peak amplitude increased with learning, from 7.4 to 11.5 Hz for the 1st and 3rd stages, respectively. Furthermore, on a single trial basis, instantaneous synchrony of topTC1 neurons was negatively correlated with lick-latency fluctuation (slope = –0.02, p<0.0001, n=2,115 trials, *Figure 5B*, see Methods for details).

Conversely, for topTC2 neurons in No-go trials, synchronous CS-triggered lick responses in the positive time domain were very strong initially, but disappeared at the later stage (peak amplitude, 7.5 and 0.7 Hz for the 1st and 3rd stages, respectively, *Figure 5C*). Regression analysis further showed that instantaneous synchrony of topTC2 neurons was positively correlated with the early lick rate in No-go trials (slope = 0.24, *P*<0.0001, n=965 trials *Figure 5D*). Note that synchronous CS-triggered lick responses of topTC1 and topTC2 neurons in the reward lick and succeeding lick windows were small and unchanged during learning, suggesting that TC1-TC2 were neither related to rewards nor their sensorimotor feedback. Moreover, multiple regression analysis of the two early lick variables (Go and No-go) with instantaneous synchrony of the four topTC neurons combined indicated that lick-latency fluctuation in Go trials was only correlated with synchrony of topTC1 neurons (*Figure 5— figure supplement 1A*), whereas the early lick rate in No-go trials was most strongly correlated with synchrony of topTC2 neurons, although moderate correlations with synchrony of topTC1 and topTC4 neurons were also observed (*Figure 5—figure supplement 1B*).

The multiple regression analysis indicated that synchrony of TC1 neurons was strongly correlated with lick-latency fluctuation in Go trials, but that of TC2 neurons was not the only determinant of the early lick rate following No-go cues. Therefore, we hypothesized that an increase in TC1 synchrony would cause a decrease in lick-latency fluctuation in Go trials and that a decrease in the fraction of TC2 neurons would cause a decrease in fraction incorrect for No-go trials. We tested these hypotheses by comparing these changes across mice. For that purpose, we first classified all recorded neurons into one of the TC1-4 populations based on their TC coefficients (see Methods). By investigating inter-subject variability (see *Figure 5— figure supplements 2 and 3* for data of individual mice), we found that the amount of change in

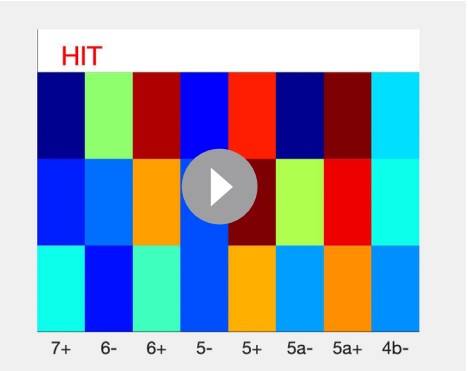

**Video 2.** Relationship of TC coefficient, synchrony strength and instantaneous synchrony within individual Ald-C compartment for HIT trials in the 1st and 3rd learning stages of TC1. For each Ald-C compartment (column), the neuron which has highest TC1 coefficients (bottom row) was selected as a reference neuron. Instantaneous synchrony in a single trial (top row) and synchrony strength (second row) were estimated between the reference neuron and other neurons in the same compartment. While synchrony strength was static within session, instantaneous synchrony varies trial-to-trial with strong values observed for HIT trials but not trials of the other cue-response conditions.
https://elifesciences.org/articles/86340/figures#video2

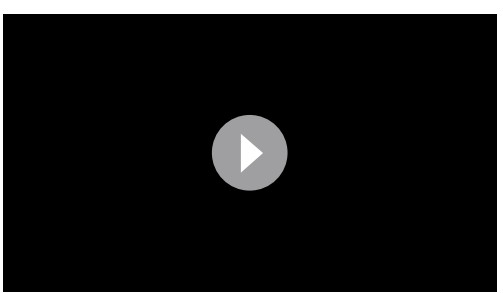

**Video 1.** CF firings in 10 ms bins of Ald-C compartment 5+neurons for HIT trials in two representative sessions of the 1st and 3rd learning stages. Detailed description for the elements can be found in *Figure 4C*.
https://elifesciences.org/articles/86340/figures#video1

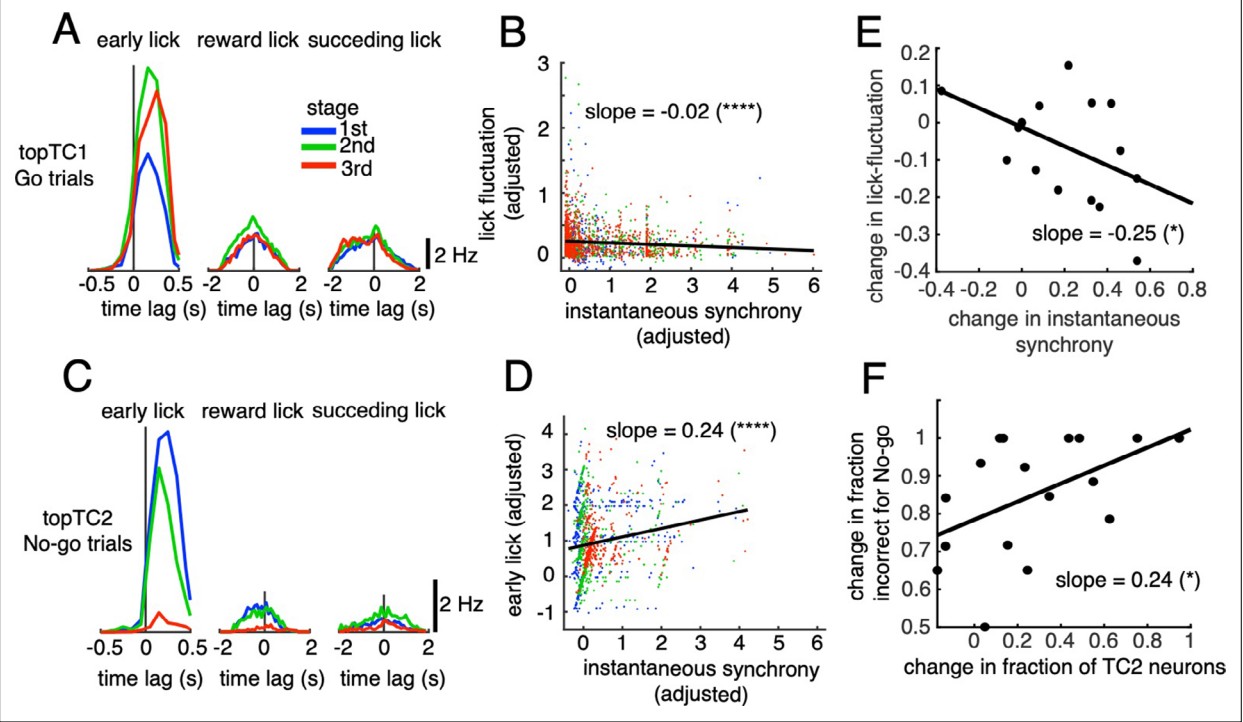

**Figure 5.** Correlations between synchronous activities in TC1-2 neurons and licking behavior. (**A**) Synchronous CS-triggered lick responses of topTC1 neurons in the three response windows: early lick (0–0.5 s after cue onset), reward lick (0.5–2 s) and succeeding lick (2–4 s). (**B**) Scatterplot of instantaneous synchrony and lick-latency fluctuation in Go trials (n=2,115) of topTC1 neurons. (**C**) Synchronous spike-triggered lick responses of topTC2 neurons in the three response windows similar to those in **A**. (**D**) Scatterplot of instantaneous synchrony and number of early licks in No-go trials (n=965) of topTC2 neurons. Each dot in scatterplots of **B-D** corresponds to a single trial. We used a multiple linear regression model with the two learning variables as functions of instantaneous synchrony of the four topTCs and fraction correct. The black trace represents the correlation of two learning variables and instantaneous synchrony, with a slope and significance level indicated by asterisks. Note that the ordinate and abscissa of scatterplots in **A-B** were adjusted to show correlations specific to topTC1 or topTC2 neurons (see Methods). (**E**) The scatter showed correlation of the amount of change in instantaneous synchrony of TC1 neurons (abscissa) and amount of change in lick-latency fluctuation (ordinate) across n=17 mice. For individual mice, the amount of change was computed as differences between sessions that have the highest and lowest fraction correct for Go cues. Each black dot represents a single animal. (**F**) Plot similar to **E**, but for the amount of change in fraction of TC2 neurons (abscissa) and amount of change in fraction incorrect for No-go cues (ordinate). The amount of changes was computed as differences between sessions that have the highest and lowest fraction incorrect for No-go cues.

The online version of this article includes the following source data and figure supplement(s) for figure 5:

**Source data 1.** Datasets used to create *Figure 5*.

**Figure supplement 1.** Multiple regression analysis for two lick variables (lick-latency fluctuation and early lick rate) and synchrony of topTC1-4 neurons.

**Figure supplement 2.** Lick-latency fluctuation in Go trials as a function of instantaneous synchrony in TC1 neurons for individual animals.

**Figure supplement 3.** Behavioral performance as a function of neuron fraction for individual animals.

**Figure supplement 4.** Decoding analysis of lick events.

**Figure supplement 5.** Effect of muscimol injection on lick timing precision.

instantaneous synchrony in TC1 neurons was significantly and negatively correlated across animals with the amount of change in lick-latency fluctuation in Go trials (slope = –0.25, p=0.02, *Figure 5E*). Similarly, the amount of change in fraction of TC2 neurons was significantly and positively correlated across animals with the amount of change in fraction incorrect for No-go cues (slope = 0.24, p=0.03, *Figure 5F*).

We further conducted a decoding analysis to test how well CS spiking predicts licking behavior. By assuming that a single spike independently triggers a lick event consistent with the CS-triggered lick histogram, we derived the likelihood of licking events, given the spike train (see Methods for details). As a result, synchronous CSs of topTC1-2 neurons predicted occurrence of lick events statistically better than all CSs of topTC1-2 neurons, all CSs of all neurons in the same recording session,

or the chance level for which no correlation between CS and lick events was assumed (*Figure 5—figure supplement 4*). At the individual trial level, synchronous CSs of topTC1 and topTC2 neurons were the best predictors for approximately 50% of all lick events in HIT and FA trials, respectively (*Figure 5—figure supplement 4C*). In addition to the decoding analysis, 4 of 5 animals showed an increase in timing fluctuation of the first-lick in HIT trials following muscimol injection at the left Crus II (*Figure 5—figure supplement 5*).

## Tensor representation in CSs of individual cells in the cerebellar cortex

While TCA clearly decomposed CSs into four TCs, we also observed overlaps in TC representation at the individual neuron level, i.e., a single neuron may represent more than one TC by having large coefficients of multiple TCs (see *Figure 4—figure supplement 1A* for the ratio of overlapping neurons among TCs). We visualized such overlap by mixing the coefficients of TC1-4 for individual neurons using CMYK colors (*Figure 6A*) and found that neurons representing different TCs could reside in the same compartments. For instance, TC1 (cyan) and TC2 (magenta) neurons could be found in AldC 6+in the first two learning stages. More strikingly, we found that a fraction of neurons represented multiple TCs. They were 'green' neurons for mixing of TC1-TC3 in AldC compartments 5a- and 5a+, and 'dark magenta' neurons for mixing of TC2-TC4 in the lateral Crus II at the first stage. The tensor representation of individual neurons showed that TCs have complicated compartmental distributions that change during learning. At the same time, single AldC compartments and single neurons can represent multiple TCs.

To systematically investigate changes in TC representation with learning, we again classified all recorded neurons as one of the TC1-4 populations based on their TC coefficients. Compartmental distributions of each TC were roughly maintained across the three learning stages as shown in *Figure 3C*. Briefly, TC1 neurons were mostly distributed in AldC-positive compartments, TC2 and TC4 neurons in the lateral Crus II, and TC3 neurons in the medial Crus II, and mainly in AldC negative compartments at the later learning stage (pie charts in *Figure 6A*). We found that fractions of TC1 and TC4 neurons increased and those of TC2 and TC3 decreased significantly (*Figure 6B*). Interestingly, while fractions of TC2 and TC4 neurons in the lateral Crus II (AldC compartments 7+to 5-) changed in opposite directions (one-way ANOVA, p<0.0001 for changes in fractions across the three stages, n=150 sessions, *Figure 6C*), their summation remained unchanged at about 60% even at the session level (ANOVA and pairwise t-tests, p>0.3), suggesting that TC2 neurons were converted into TC4 neurons during learning.

Next, we constructed a multiple linear regression model to evaluate simultaneous dependence of synchrony strength on geographic distance and TC distance (difference in TC coefficients) of given pairs of neurons (see Methods for details). Note that this regression model is practically plausible, thanks to a relatively weak correlation between the two explanatory variables (geographical distance vs. TC distance, correlation coefficient $R$=0.14, n=170,396 neuronal pairs). We found that the synchrony strength was significantly and negatively correlated with both geographic distance (slope = –0.0004, p<0.0001, *Figure 6D*) and TC distance (slope = –0.07, p<0.0001, *Figure 6E*). These results suggest that geographically close neurons tend to be more synchronized than distant ones, and that neurons with similar TC coefficients tend to be more synchronized than those with different TC coefficients, even when canceling out the other factor.

We further investigated how boundaries of AldC expression determine synchrony strength in addition to geographic and TC distances among neurons, by pooling neuronal pairs into within- (n=87,192 pairs) and across-compartment (n=83,204 pairs) groups. We found that synchrony strength was higher for the within-compartment group (mean ± std, 0.18±0.12) than for the across-compartment group (0.13±0.09, one-way ANOVA p<0.0001, *Figure 6F*). As expected, both geographic distance and TC distance were lower for the within-compartment group than for the across-compartment counterpart (geographic distance, 43±31 and 114±53 µm; TC distance, 0.30±0.31 and 0.39±0.33 for within-compartment and across-compartment groups, respectively, p<0.0001 for all pairwise comparisons using one-way ANOVA).

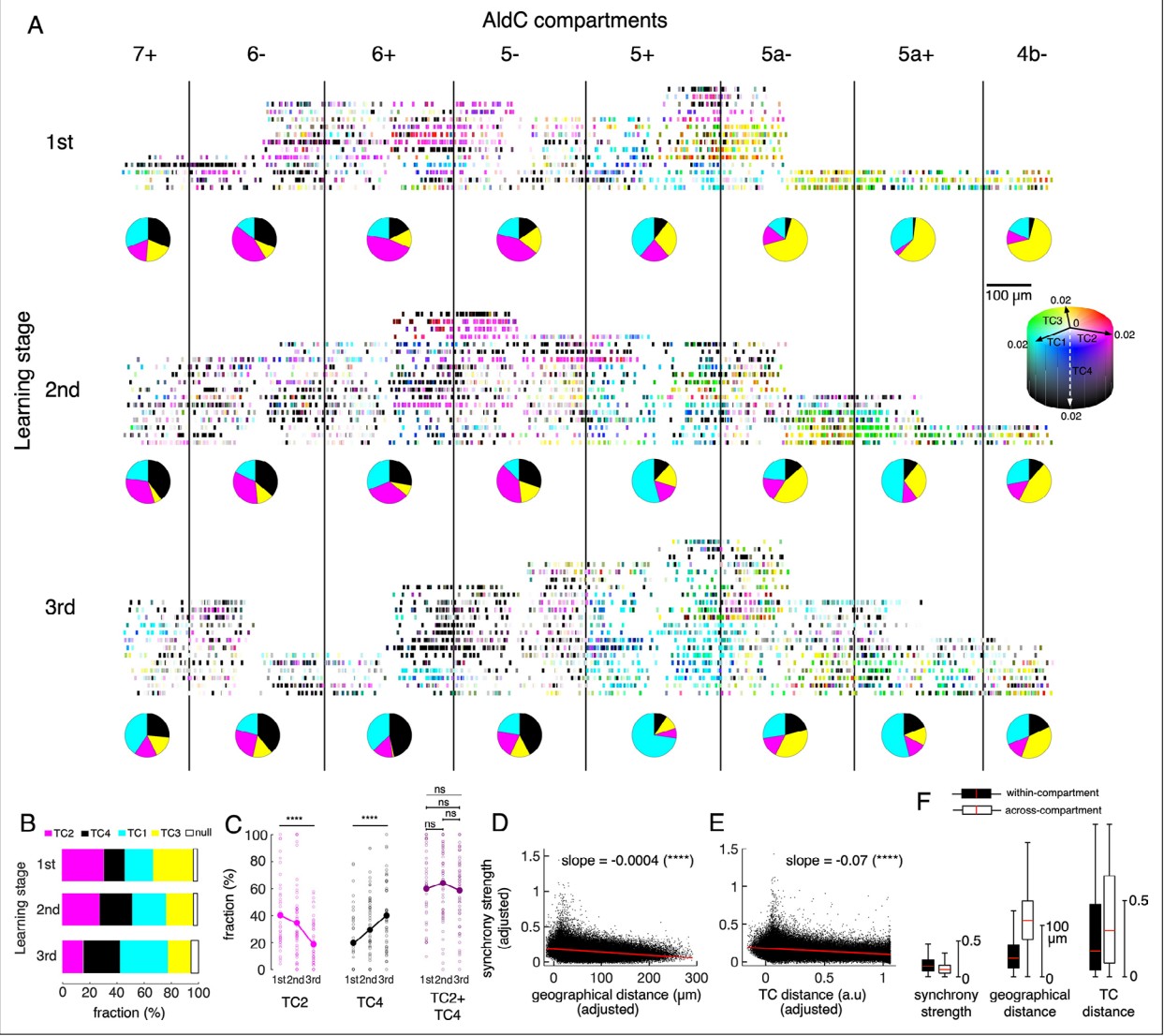

**Figure 6.** Tensor representation of individual Purkinje cells in Crus II. (**A**) CS firing in n=6,465 neurons in 8 AldC compartments (columns) and at three learning stages (rows) were evaluated by TCA. Each short bar indicated the location of a single cell relative to the AldC boundaries (vertical black lines). The color of each cell was mixed by coefficients of the four TCs (cyan – TC1, magenta – TC2, yellow – TC3 and black – TC4). Each row corresponds to a single recording session. For visualization purposes, the width of each AldC compartments was manually adjusted to 300 µm. Pie-charts indicate fraction of TC1-4 neurons in each AldC compartment and at each learning stage. (**B**) Fractions of neurons classified as TC1-4 in each of the three learning stages (color bars). Note that less than 6% of all recorded neurons could not be classified as TC1-4 (null, open bars). (**C**) Fractions of neurons in the lateral Crus II (AldC compartments 7+, 6-, 6+, 5-) classified as TC2 (magenta) and TC4 (black) and their summation (TC2 + TC4, dark magenta) in n=150 sessions (small open circles) and their means (solid large circles) for the three learning stages. (**D–E**) Correlations between the synchrony strength vs. geographical distance (**D**), synchrony strength vs. TC distance (**E**). A single black dot corresponds to a neuronal pair (n=170,396 pairs). The red line represents the correlation between measures with a slope and significance level indicated by asterisks. Note that the ordinate and abscissa of scatterplots in **D**-**E** were adjusted to show correlations specific to geographic distance or TC distance between neuron pairs, similar to the multiple regression analysis performed in *Figure 5B&D*. (**F**) Box plots show the three measures sampled for within-compartment (filled boxes, n=87,192 pairs) and across-compartment groups (open boxes, n=83,204 pairs). Description of the box plot is the same as in *Figure 2E–I*.

The online version of this article includes the following source data and figure supplement(s) for figure 6:

**Source data 1.** Datasets used to create *Figure 6*.

**Figure supplement 1.** PCA of Go/No-go data.

**Figure supplement 2.** Possible functions of TC3 and TC4.

## Discussion

The significance of our work is calcium imaging of CF inputs spanning the entire dorsal surface of Crus II during a reward-driven Go/No-go learning task. This allowed us to study functional differences between Aldolase-C compartments in a single task that required various cerebellar functions. Our previous study (*Tsutsumi et al., 2019*) suggested that distinct characteristics of CF inputs can be presented simultaneously in multiple cerebellar compartments. However, because analyses in that study were performed for calcium responses with a relatively low temporal resolution (less than 10 Hz) and after the learning is completed, that is, after mice reach the expert level, it remained unclear which neural mechanisms are important and how they are organized as cerebellar components as learning progresses. In the present study, we expanded the analysis of CF inputs throughout the course of learning. Furthermore, timing of complex spikes (CSs) was estimated at 100 Hz resolution using the hyperacuity algorithm (*Hoang et al., 2020b*). Then trial-averaged CS activities were decomposed using tensor component analysis (*Williams et al., 2018*). Three main new findings were drawn based on those two technical advances. First, we found that all CS activity can be decomposed into only four components (TC1-4), which capture key features of behavior and learning. This is a significant dimension reduction performed by the cerebellum. We also found that compartmental representations of these populations align well with functional and anatomical boundaries between medial and lateral parts of Crus II, as well as expression of Aldolase-C. Second, CS synchrony, measured at single-trial level, was concentrated among neurons that belong to the same components. Furthermore, the synchrony strength in different components may change in the opposite direction during learning, suggesting that CS synchrony is the neural mechanism that drives organization of those components. Third, we suggested that, increase of CS synchrony in TC1 neurons may behaviorally decrease the lick fluctuation in Go trials across animals. In parallel for cognitive learning, a decrease of the relative proportion of TC2 neurons may decrease cognitive error of No-go cues across animals. Together, these results indirectly imply causal relationships of TC1 with timing control and TC2 with cognitive error learning. In summary, our study suggested that bi-directional synchronous response-associated changes in CF activities, finely constructed on compartmental structure, could reduce the dimension of the learning space and thus provide a flexible learning scheme in diverse cerebellar functions.

### Multiple cerebellar components in a single task

One of the valuable features of this study was that we applied the hyperacuity algorithm (*Hoang et al., 2020b*) to estimate CS timings from calcium responses with 10 ms resolution. This enabled us to study a huge number of neurons (>6000) with hyperacuity resolution (10ms) along a whole process of cognitive and motor learning, with precise spatial localization guided by boundaries between Aldolase-C (AldC) zones. Tensor component analysis (TCA) of this high spatio-temporal resolution CS data revealed four distinct components that may be involved in different learning-related functions (see Methods and *Figure 6—figure supplement 1* for principal component analysis of the same CS data). Notably, the first two components are in good agreement with the dichotomy in the literature that CFs either function as "error signals" or "timing signals" (*Mauk et al., 2000*). In particular, the TC1 population consisted of neurons in AldC-positive zones with fast responses in the HIT condition (*Figure 3A*). During learning, increased CS synchrony in TC1 neurons was correlated with the decrease of lick-latency fluctuation (*Figure 5B*). Furthermore, spike-triggered lick responses of TC1 neurons were almost zero for negative time lag and sharply rose in positive time lag (*Figure 5A*). Strikingly, analysis of inter-subject variability across 17 mice showed that the change in CS synchrony of TC1 neurons was negatively correlated with change in lick-latency fluctuation in Go trials (*Figure 5E*). These results suggest that CFs projecting to TC1 neurons convey "timing signals" to control early licks, in accordance with timing control hypothesis (*Lang et al., 1999*; *Welsh et al., 1995*; *Wagner et al., 2021*; *Llinás, 2013*; *Ohmae et al., 2017*). That is, increased CS synchrony could stabilize the timing of motor commands by canceling noisy synaptic inputs to the IO, or contributing to timed motor initiation by evoking rebound firing in downstream cerebellar nuclei (*Tsutsumi et al., 2020*; *Tang et al., 2019*), possibly leading to more precise timing control with less fluctuation. In contrast, the TC2 population, distributed mostly in the lateral Crus II, showed very fast responses in the FA condition (*Figure 3A*). As learning proceeded, TC2 synchronous firings in the FA condition decreased dramatically (*Figure 4E*) and such a decrease was highly correlated with a decrease in the early lick rate (*Figure 5D*). Furthermore, the change in the relative proportion of TC2 neurons was positively correlated with the change

in the fraction incorrect of No-go cues across animals (*Figure 5F*). These results suggested that TC2 neurons convey cognitive "error signals" (unwarranted licks) specific to the No-go cue, in accordance with the Marr-Ito-Albus hypothesis (*Kawato et al., 2021*; *Marr, 1969*; *Albus, 1971*; *Ito, 1970*). Together with decoding analysis (*Figure 5—figure supplement 4*) and muscimol injection (*Figure 5—figure supplement 5*), all of those results suggested causal relationships of TC1 with timing control and TC2 with cognitive learning.

In contrast to TC1 and TC2 populations that exhibited marked learning effects, the TC3 population showed only slight changes in CF firing and synchrony, except for becoming silent at the final learning stage for the FA condition (*Figure 4—figure supplement 1C*). The correlation between CSs of TC3 neurons with reward licks was high for HIT trials across all learning stages (*Figure 6—figure supplement 2A*), suggesting the role of TC3 in reward processing. We also found that TC4 is only related to CR trials (*Figure 3B*). Increased CS firing in TC4 neurons with learning, especially in AldC compartment 6+, led to the hypothesis that TC4 induces more successful suppression of early licks after the No-go cue, that is, an increase of CR trials and decrease of FA trials (*Figure 6—figure supplement 2B*). Spike-triggered lick responses supported this idea; TC4 firings were correlated with suppression of licks following No-go cues compared to those in the pre-cue period (*Figure 6—figure supplement 2C*).

From a conceptual standpoint, the fact that motor, cognitive and reward elements are simultaneously represented by the four TCs suggest a new direction for future investigations of CF activity. That is, populations of PC populations may optimize distinct sub-goals using their own neural mechanisms, but they may collectively serve a single task. To observe these phenomena, the task should be complex enough to include several behavioral elements requiring different PC populations, although the importance of these elements may vary depending upon the task. For instance, we insisted that learning in TC2 neurons is more important for the cognitive aspect of the present Go/No-go tasks, because the reward acquisition is determined by correct cue discrimination, while improved motor skills by learning in TC1 neurons would only maximize the reward amount per time. In eye-blink conditioning, which may contain similar cognitive and motor elements, timing control is more crucial than cue recognition because the conditioned response (e.g. closing the eyelid) requires more temporal precision.

## Dimension reduction in the cerebellum

From a computational learning-theory point of view, it is striking and extremely important that only four components seem to account for a wide variety of CF responses of Purkinje cells in eight AldC compartments. We found that four TCs accounted for more than 50% of the variance of 6,445 neurons in four cue-response conditions, while about 20 TCs are needed to explain the same amount of variance for shuffled data (*Figure 3—figure supplement 1A*). Consequently, significant dimension reduction (fivefold) was demonstrated by the permutation test. It is worth noting that these four components also explain main learning effects in the Go/No-go experiment, that is, more precise timing control, error decrease, reward processing, and successful lick suppression. Dimension reduction was further confirmed by conventional principal component analysis (PCA) showing d=67 dimensions (*Mazzucato et al., 2016*), significantly smaller than >6000 recorded neurons, explained 65% of variance (*Figure 6—figure supplement 1A*). Such dimension reduction by the cerebellum in Go/No-go tasks is remarkable compared to other brain regions in various tasks, although dimension reduction may depend on complexity of the task (*Gao et al., 2017*). Considering various factors involved in learning, such as auditory sensory inputs to the cerebellum representing different frequencies, relevant deep cerebellar nucleus (DCN), timing of licks with respect to auditory cues, and rewards, degrees of freedom of the Go/No-go tasks are huge. If a learning system possesses many degrees of freedom, a huge training dataset is theoretically required. But mice learned auditory discrimination Go/No-go task within hundreds of trials. This implies that the cerebellum reduces the degrees of freedom significantly, thanks to its zonal structure as well as effective mechanisms of dimension control by upstream networks such as gap-junctions in IO neurons (*Hoang et al., 2020a*). Dimensional reduction is certainly beneficial for the whole brain during learning, by facilitating the coordination between the cerebellum, the cerebral cortex, and the basal ganglia in different learning stages (*Bostan and Strick, 2018*; *Wagner and Luo, 2020*; *Kawato and Cortese, 2021*; *Kostadinov and Häusser, 2022*; *Wagner et al., 2019*).

## CS synchrony drives dynamic organization of functional components

We found that CS synchrony was significantly stronger within TCs than across TCs, and the highest synchrony was observed when TCs function in their dedicated cue-response conditions (*Figure 4A–C*). While CS synchrony was dependent on AldC expression and geographic distance, as already reported (*Hoogland et al., 2015*; *Tsutsumi et al., 2015*), our analysis further showed that synchrony was stronger for pairs of neurons with smaller difference in their cue responsiveness, estimated by the TC distance (*Figure 6E–G*). Together, these results suggested that synchronization, partially guided by anatomical structure, dynamically organizes diverse functions represented by TCs.

Remarkably, we also found that CS synchrony is spatially and temporally dynamic, in concert with the organization of TCs. That is, synchronous firing decreased in TC2 neurons for the FA, but it increased in TC1 neurons for the HIT condition (*Figure 4E*). Such opposite changes in CS synchrony were strongly correlated with changes in cue responsiveness (*Figure 4* and *Figure 2—figure supplement 2C*). This phenomenon raised an important question of which factor, CSs or their synchrony, drives licking behaviors. Our decoding analysis indicated that synchronous CSs better predict licking events than ordinary CSs (*Figure 5—figure supplement 4*). Furthermore, previous studies showed that highly synchronous CSs caused a large drop in DCN activity while isolated CSs produced only weak inhibition (*Tang et al., 2019*; *Tang et al., 2016*). These results together supported the view that synchronous CS activity shapes DCN output generating motor commands. If that is true, why does CS synchrony change in an opposite direction among different neuronal populations, for example TC1 vs. TC2? We have two plausible explanations for this. One mechanistic explanation could be different synaptic plasticity rules between AldC-positive and -negative zones. Note that TC1 is mainly distributed in AldC-positive zones (*De Zeeuw, 2021*). Another functional hint may come from desired outputs of cerebellar coding in a particular process. Enhanced CS synchrony in TC1 would induce synchronous activation of Purkinje cell ensembles, which in turn, would tune downstream systems to facilitate initiation and coordination of precise timing control of movement. In contrast, strong CS synchrony in TC2 neurons at the initial learning stage would reduce the dimensionality of high-dimensional cognitive error signals for fast learning. At later stages, low CS synchrony would increase the dimensionality of cerebellar output for fine and sophisticated learning (*Kawato et al., 2011*; *Tokuda et al., 2013*). This argument was supported by the result showing that dimensions of CS firings in TC1 and TC2 neurons decreased and increased, respectively, in accordance with changes in their CS synchrony (*Figure 3—figure supplement 1D, E*). These opposite changes in the dimension could be realized by interaction between electrical synapses in the IO and positive feedback loop of PC, DCN, and IO, but future experiments are needed to test these possibilities.

## Duality of motor control and cognitive learning in TC2 neurons

Beside their correlation with error signals, synchronized CSs of TC2 neurons may also convey motor commands to control early licks in No-go trials (*Figure 5C*). Such a duality of motor control and cognitive learning has never been reported, despite a mixed representation of error information and motor-related variables, for example direction of movement, in CSs was observed in hand movement (*Kitazawa et al., 1998*; *Sendhilnathan et al., 2021*; *Ikezoe et al., 2023*) and saccade (*Markanday et al., 2021*). This finding implies that synchronized CSs of TC2 neurons do not simply multiplex complementary behavioral information. Rather, they contribute to generation of motor commands in downstream systems to inhibit the licking following No-go cues. One possible interpretation is that PCs, which are involved in cognitive learning, are assumed to modulate simple spikes (SSs) that also induce licks in FA trials. Long-term depression (LTD) may occur by co-activation of parallel fibers and CF inputs to these PCs (*Ito and Kano, 1982*; *Wang et al., 2000*; *Doi et al., 2005*; *Jörntell and Hansel, 2006*). As a result, simple spikes (SSs) tend to be modulated so that fewer erroneous licks are generated. We note that suppression of SSs may drive the movement in other learning schemes. For instance, in eye-blink conditioning, SS suppression is correlated with conditioned blink responses on a trial-by-trial basis in awake mice (*ten Brinke et al., 2015*).

It is interesting that TC2 and TC4 neurons (both responded actively to the No-go cue) possess not only almost identical temporal response profiles and overlapping spatial distributions (*Figure 3*), but also the largest overlap among TCs (*Figure 4—figure supplement 1A*). Furthermore, although fractions of TC2 and TC4 neurons decreased and increased significantly, summation of their fractions remained unchanged during the course of learning (*Figure 6D*). We hypothesized that TC2 and TC4

neurons are of the same neuron population, but that they changed their cue-response specificity (FA to CR) due to LTD of parallel-fiber-Purkinje-cell synapses guided by CF inputs (*Yamamoto et al., 2002*). Future simultaneous recordings of SSs and CSs as well as downstream systems may reveal neural mechanisms of TC2 and TC4 neurons.

### CFs multiplex various information

Limitations of the current study are that activities of the four TC populations described above were of neurons sampled in different recording sessions, and that we did not investigate in detail single populations throughout learning. Thus, monitoring population activities as learning proceeds, combined with causal analysis of neuronal responses and behavior changes, is needed to understand diverse CF functions across TC populations. This is particularly crucial because CFs may multiplex motor, cognitive and reward-related information (*Kitazawa et al., 1998*; *Sendhilnathan et al., 2021*; *Ikezoe et al., 2023*; *Markanday et al., 2021*). In *Figure 6*, we highlighted this possibility by showing a mixed tensor representation of 6445 individual neurons in the cerebellar cortex. Even though each TC has a unique zonal distribution, individual zones or even neurons may represent multiple TCs and may be involved in different cerebellar functions. However, unlike previous studies, multiplexing of CFs in the present study was derived solely from TC coefficients of the neurons. A formal encoding analysis in future work may reveal what variables single CFs multiplex in the Go/No-go task.

## Methods

All experiments were approved by the Animal Experiment Committees of the University of Tokyo (#P08-015) and the University of Yamanashi (#A27-1), and carried out in accordance with national regulations and institutional guidelines. Seventeen adult male heterozygous Aldoc-tdTomato mice (*Tsutsumi et al., 2015*) and adult male wild-type mice (Japan SLC, Inc, n=5) at postnatal days 40–90 were used. The Aldoc-tdTomato mouse line is available from the corresponding authors upon request, and mice are also available at the RIKEN BioResource Center (RBRC10927).

### Surgery and two-photon recordings

Cranial window surgery and two-photon recordings in the cerebellum of mice during the Go/No-go auditory discrimination task were conducted in the same way as reported in *Tsutsumi et al., 2019*. Briefly, a cranial window was created over the left Crus II of mice anesthetized with isoflurane (5% for induction and 2% for maintenance), and adeno-associated viruses encoding GCaMP6f was injected. After the recovery period, mice were acclimated to head-fixation and trained to perform the Go/No-go auditory discrimination task. Two-photon imaging was performed during the following Go/No-go task phase at a scanning rate of 7.8 Hz, using a two-photon microscope (MOM; Sutter Instruments) equipped with a 40 x objective lens (Olympus) controlled by ScanImage software (Vidrio Technologies). Imaging data were analyzed using MATLAB (R2018a; MathWorks). Two-photon recording experiments were conducted once daily for 4–7 consecutive days, for a total of 236 sessions, each containing 13–43 trials. The inter-trial interval was fixed at 6 s, but the subsequent trial was delayed by 1 s from the last lick if the mouse continued to lick. Liquid rewards were given three times with an interval of 0.41 s (0.41, 0.82, and 1.23 s) after timing of the first lick.

### Two-photon recording data during the Go/No-go auditory discrimination task

Some two-photon recording data, that is after mice reach expert level, were previously analyzed (*Tsutsumi et al., 2019*). Here, we analyzed all recorded neurons during the course of learning. Each trial with two-photon recording data was categorized as HIT, FA, CR, or MISS, according to licking behavior within a response period of 1 s to the two cues, that is correct lick in response to the go cue, unwarranted lick in response to the No-go cue, correct response rejection to the No-go cue, or response failure to the Go cue, respectively. Each session was also categorized according to the fraction correct of mouse performance, that is the ratio of correct responses (HIT and CR) to all trials of the session. Categories included the 1st, 2nd, and 3rd stages of learning with the fraction correct <0.6, 0.6–0.8, and >0.8, respectively. The fraction correct for the Go cue was evaluated as the ratio of HIT trials to Go trials. The fraction incorrect for No-go cue was evaluated as the ratio of FA trials to No-go

trials (*Figure 1C*). For each session, on average, there were approximately 30 neurons simultaneously recorded while more than 30 trials of Go/No-go cues were randomly presented. In total, there were 59, 83, and 94 recording sessions, 1462, 2405, and 2578 neurons and 1552, 2731, and 3692 trials in the 1st, 2nd, and 3rd stages, respectively (see *Supplementary file 1* for more details).

## Calculation of licking variables associated with cues

We evaluated two licking variables associated with learning of Go and No-go cues in the early response window (0–0.5 s after cue onset). Lick-latency was estimated as the difference in timing of the first lick onset and the cue onset. We found that during learning, there were no systematic changes in the mean of lick-latency, but its fluctuation from the mean was significantly reduced (*Figure 1—figure supplement 1A*). To account for slight variations in lick-latency across training sessions, we fitted the lick-latency as a function of the trials, sorted by training session, by a $4^{th}$ order polynomial curve for individual mice, which is the best model among 0 to 5th order polynomial models with Akaike Information Criterion. The lick-latency fluctuation of a single trial was then calculated as the absolute difference between the lick-latency and the fitted curve, normalized by the mean of lick-latency for individual mice (*Figure 1—figure supplement 1B*). Lick-latency fluctuation was computed only for HIT and FA trials, omitting MISS and CR trials, for which there were no licks in the early response window. The early lick rate was counted as the number of licks in the early response window. For each animal, we fitted lick-latency fluctuation in Go and number of early licks in No-go trials using two linear models of trials, respectively; hence, their slopes indicated the corresponding learning effects, that is negative slopes for less fluctuation in lick-latency after Go cue and less erroneous licks after No-go cue.

## Reconstruction of spike events

$Ca^{2+}$ signals corresponding to CF inputs were evaluated for PC dendritic regions-of-interest (ROIs) manually selected from the candidates extracted by Suite2p software (*Pachitariu et al., 2017*). Spike trains were reconstructed for 6,445 Purkinje cells sampled in the 17 mice, using hyperacuity software (*Hoang et al., 2020b*) (HA_time) that detected spike activities for $Ca^{2+}$ signals of two-photon imaging with a temporal resolution of 100 Hz. The mean CF firing rate (1.1±0.4 spikes/s) and cross-correlograms (CCGs) of CFs across neurons within individual Ald-C compartments were consistent with those for previous studies using electrical recordings in behaving (*Tsutsumi et al., 2020*) and anesthetized (*Blenkinsop and Lang, 2006*) mice (*Figure 2—figure supplement 1B, C*). Furthermore, simulations of observed $Ca^{2+}$ signals using GCaMP6f dye showed that the HA_time was capable of detecting roughly 90% of the spikes (*Figure 2—figure supplement 1D*). Together, these results guaranteed the reliability of HA_time in detecting complex spike timings with high temporal precision. We note that to compensate for small jitters of spike timing estimation as well as to increase the signal-to-noise ratio, we used 30 ms bins to evaluate synchrony and 50 ms bins to construct peri-stimulus time histograms of CS activity (see below).

## CS responsiveness to cue stimulus

To evaluate CS responsiveness of AldC compartments to cue stimuli, we constructed peri-stimulus time histograms (PSTH) of CS activity with a time bin of 50ms. PSTHs were constructed for individual Purkinje cells and averaged across Purkinje cells in the same AldC compartment (*Figure 2A–D*). We also evaluated response strength as CS firing rate in 0–200ms after cue onset, since the peaks of PSTHs were mostly 200–300ms. To demonstrate opposite response changes between HIT and FA of medial and lateral parts of Crus II, for each neuron, we computed the difference in response strengths between HIT and FA trials (i.e. HIT - FA). We then selected the top 100 neurons in each AldC 5+and 6- with highest and lowest values, respectively.

### Tensor component analysis

Since most CSs reduce their activity to baseline 2 s after the auditory cue, we conducted tensor component analysis (TCA) (*Williams et al., 2018*) on PSTHs sampled from –500ms to 2 s from cue onset (bin size, 50ms) of all Purkinje cells (n=6,445) in the four cue-response conditions, i.e., HIT, FA, CR, and MISS. Each PSTH was subtracted from its baseline activity, defined as the mean of PSTHs in the range of [-2,–1] s before cue onset. If the number of trials of a specific cue-response condition in a recording session was less than 5, we fixed corresponding PSTHs of the neurons in that session to 0.

Let $x_{ntk}$ denote the PSTH of neuron $n$ at time step $t$ within cue-response condition $k$. TCA yields the decomposition

$$x_{ntk} \approx \hat{x}_{ntk} = \sum_{r=1}^{R} \lambda_r w_n^r b_t^r a_k^r$$

, where $R$ is the number of tensor components, $w_n^r$, $b_t^r$ and $a_k^r$ are the coefficients of the neuron, temporal, and response condition factors, respectively. Those coefficients were scaled to be unit length with the rescaling value $\lambda_r$ for each component $r$. We introduced a non-negative constraint of those coefficients ($w_n^r \geq 0$, $b_t^r \geq 0$ and $a_k^r \geq 0$ for all $r$, $n$, $t$ and $k$). *Figure 3* showed the coefficients $w_n^r$, $b_t^r$ and $a_k^r$ for each of the tensor component $r$=1,2,3,4.

TCA iteratively estimated the coefficients with an alternating least-squares algorithm; thus, its results are dependent on random initial values. The number of tensor components was systematically examined with $R$=1–20, each with 100 random initializations. For each $R$, we selected the optimal solution as the one that was obtained most frequently among 100 initializations. To evaluate the fitting performance, original PSTHs, $x$, and those reconstructed from TCA coefficients, $\hat{x}$, 0–1 s after cue onset were first low-pass filtered (cut-off frequency of 2 Hz). Then variance accounted for (VAF) was computed as follows

$$\text{VAF} = 1 - \frac{var\left(x - \hat{x}\right)}{var\left(x\right)}$$

For each value of $R$, we further inspected the similarity of the optimal solution and the solutions obtained from the other 10 random initializations. We selected $R$=4 which accounted for more than 50% of total variance and provided stable solutions (see *Figure 3—figure supplement 1A–B*). Increasing $R$=1–3 to 4 added meaningful components, that is segregation of TC4 from TC2 specific for CR trials. However, further increasing $R$ separated the components with slight differences observed only in temporal profiles (*Figure 3—figure supplement 1C*).

We defined the TC distance of the two neurons using cosine distance:

$$\text{TC distance}_{i,j} = 1 - \frac{w_i w_j'}{\sqrt{\left(w_i w_i'\right)\left(w_j w_j'\right)}}$$

where $w_i$ and $w_j$ are the neuronal coefficients of the $i$-th and $j$-th neurons, respectively.

We also applied TCA for the shuffled data, in which the firing activity of all neurons and in all cue-response conditions was permuted. For each permutation, the firing activity of a neuron in a particular trial was taken from that of the other trial randomly selected and of the neuron randomly selected. This process was repeated for all 6,445 neurons and all recording trials. Note that while the firing activity was shuffled, the labeling of the newly inserted trial regarding cue-response condition was relabeled as the same as that of the original data (*Figure 3—figure supplement 1A*). Following this shuffle, the shuffled data had exactly the same structure as the original data. Then we estimated PSTHs for the shuffled data and conducted TCA in the same way as described above. The null hypothesis is that the firing activity is independent of the cue-response condition, and thus shuffling spiking data across trials and neurons does not affect the results of TCA. We shuffled the data 100 times and compared their VAF profiles with that of the original data (*Figure 3—figure supplement 1A*).

## Sampling neurons by TCA coefficients

For further analyses of the four tensor components retrieved by TCA, we sampled neurons that were best represented by each component as follows. First, at each learning stage (1st - 3rd stages), we selected the top 300 neurons that had the largest coefficients for each of the four components. Then, neurons that were sampled by more than one component( i.e. that overlapped) were excluded (*Figure 4—figure supplement 1A*). As a result, we sampled 2,096 neurons for the four tensor components. These sampled neurons (abbreviated by 'topTC neurons') were analyzed in *Figure 4* and *Figure 5A–D*. Note that because different numbers of neurons were recorded across learning stages,

we also sampled the top 10–20% of TC neurons using the above process and found identical PSTHs of TCs (*Figure 4—figure supplement 1C*).

## Classification of all recorded neurons by TCA coefficients

For *Figure 5E–F* and *Figure 6*, which analyzed all recorded neurons, neurons were classified into one of the TC1-4 populations based on their TC coefficients, for example a particular neuron is classified as TC1 if its coefficient of TC1 is the highest among four TC coefficients. The neurons, whose TC coefficients were all zero, were considered as null.

## Spike-triggered lick response

To investigate the correlation of CF activity and licking behavior, we sampled spikes and lick onsets in the three windows of each trial: 0–0.5 second (early lick), 0.5–2 s (reward lick) and 2–4 s (succeeding lick) after cue onset. Then, the spike-triggered lick response was constructed across trials with a time bin of 100ms, normalized by the total number of trials. We note that for TC1 and TC2 (c.f. *Figure 5*), we examined lick responses for synchronous spikes, those that were co-activated (in the time bin of 30ms) with at least one spike of the other neurons of the same TC in the same recording session. For TC3 and TC4, since their synchrony changes were modest (*Figure 4—figure supplement 2*), we examined all sampled spikes (*Figure 6—figure supplement 2*).

## Synchrony analysis of CF activity

In this study, we evaluated synchronization of CF activities by two indices, one on a trial basis (we named it 'instantaneous synchrony') and the other across trials in the same recording session ('synchrony strength'). For the former, we estimated the instantaneous synchrony in each trial by the total number of synchronous spike pairs (co-activated in time bins of 30ms) in the window of 300ms before the first lick, normalized by the number of cell pairs. Suppose 10 neurons were simultaneously measured and together they produced 20 spikes within the time window. Further suppose 4 neurons are co-activated in one time bin and 2 neurons were co-activated in another time bin (the other 14 spikes fired in separated time bins). Then instantaneous synchrony is calculated as $\frac{C_4^2+C_2^2}{C_{10}^2} = 0.15$. For this example, the spike count is 20/10=2. We note that for CR and MISS trials, for which no early lick was generated, the time window was fixed as 0–300ms after cue.

We also measured synchrony strength across trials in two different neurons by the cross-correlograms (CCGs). The spike train of a neuron was represented by $X(i)$, where $i$ represents the time step ($i=1, 2, …, N$). $X(i)=1$ if spike onset occurs in the $i$-th time bin; otherwise, $X(i)=0$. $Y(i)$ was the same as $X(i)$, but for the reference neuron. The cross-correlation coefficient at time lag $t$, $C(t)$, was calculated as follows:

$$C(t) = \frac{\sum_{i=1}^{N} X(i) Y(i - t)}{\sqrt{\sum_{i=1}^{N} X(i) \sum_{i=1}^{N} Y(i)}}$$

A 10 ms time bin was used; thus, for two spikes to be considered synchronous, their onsets must occur in the same 10 ms bin. Synchrony strength was defined as the sum of $C(t)$ in a window of ± 10ms around the zero-lag time bin $C(0)$. Cross-correlation caused by spike synchronization to the task event stimulus was evaluated as a CCG, for which the spike time of the reference neuron was shifted by one trial period (*Perkel et al., 1967*; *Toyama et al., 1981*).

## Regression analysis of synchrony and licking variables

We fitted a multiple linear regression model on a trial basis, with the synchrony of each TC as an explanatory variable and two licking variables as response variables. The formula of this model specification was shown by Wilkinson notation as

$$y \sim syn_{\text{TC1}} + syn_{\text{TC2}} + syn_{\text{TC3}} + syn_{\text{TC4}} + fraction\ correct$$

in which $y$ was either lick-latency fluctuation or the number of early licks, $syn_{\text{TC1-TC4}}$ were synchrony of TC1-4 neurons of the same trial, and *fraction correct* was common across trials in the same recording session. *fraction correct* was introduced to represent general learning effects, except neural synchrony. Note that the variable $syn_{\text{TC1-TC4}}$ of a particular trial was set to 0 if there were no selected neurons as

TC1-4 in that trial. For example, suppose 2 and 4 neurons are selected as TC1 and TC2 neurons for a single trial, respectively. Then $syn_{TC1}$ and $syn_{TC2}$ are computed as the number of synchronous spikes of 2 and 4 neurons for that trial, respectively, and $syn_{TC3}$ = $syn_{TC4}$=0. Due to the neural sampling process, there existed 2,465 trials for which none of the neurons was selected by any of the four TCs. We excluded those trials from the analyses. As a result, multiple regression was conducted for 3,080 trials. We note that the linear mixed-effects model with random effects for intercept grouped by animal, i.e, $y \sim syn_{TC1} + syn_{TC2} + syn_{TC3} + syn_{TC4} + fraction\ correct + (1|animal)$, with animal = 1..17 as the mouse index, showed little difference from the above model, indicating that there was no across-mice effect.

We constructed added variable plots, in which variables were adjusted for visualizing partial correlations between licking response variables and an individual explanatory variable (predictor) conditional on other explanatory variables (*Figure 5* and *Figure 5—figure supplement 1*). Adjusted values are equal to the average of the explanatory variable plus the residuals of the response variable reconstructed by all explanatory variables except the selected explanatory variable. Note that the coefficient estimate of the selected predictor for the adjusted values is the same as that in the full model including all predictors. Multiple regressions and added variable plots were conducted using MATLAB functions, *fitlm* and *plotAdded*, respectively. The slope (coefficient) was reported with a significant p-value of the t-statistic for a two-sided test with the null hypothesis that the coefficient is zero.

## Examination of the hyperacuity algorithm (HA_time) by simulations

We examined HA_time by simulating calcium responses that reproduce those observed in two-photon recordings. In particular, spike events were generated according to a Poisson distribution with a mean firing rate of 1 Hz. Calcium responses were simulated by convolving the double exponentials with the spike events. The rise time constant was fixed at 10ms, while the decay time constant was 300ms corresponding to that of the GCaMP6f dye. Gaussian noise was added to reproduce the SNR (SNR = 10 dB) of experimental data. A total of 5 cells each with 100 spikes were generated for testing HA_time performance (*Figure 2—figure supplement 1*).

For performance evaluation, a correct hit case was defined as one in which the time difference between an estimated spike and a true one was smaller than the sampling interval (100ms in simulations). This process was repeated to find all hit cases between two spike trains. Remaining spikes in the true spike train were counted as missed spikes while spikes remaining in the estimated spike train were false-positives. We used the f1-score, the arithmetic mean of the sensitivity and precision, to evaluate spike detection performance.

$$\text{sensitivity} = \text{hit} / (\text{hit} + \text{miss})$$

$$\text{precision} = \text{hit} / (\text{hit} + \text{false positive})$$

$$\text{f1-score} = 2 \times (\text{sensitivity} \times \text{precision}) / (\text{sensitivity} + \text{precision})$$

## Decoding analysis

We conducted a decoding analysis to test whether synchronous spikes of TC1-2 neurons could predict the occurrence of lick events in the 0–1 s window better than the other three models including all spikes of TC1-2 neurons, all spikes from all neurons in the same recording session, and a "chance" model for which there was no spike-lick correlation.

We first constructed a common spike-triggered lick response for all spike and lick events in HIT and FA trials (*Figure 5—figure supplement 4A*). From a probabilistic view-point, this spike-triggered lick response could be considered as the probability of a lick event $l$ given a single spike $s$ in the same trial, $p(l \mid s)$. With an assumption that a single spike caused a lick independently, the probability of a lick event given a spike collection $S^M$ of the model $M$ follows:

$$p\left(l \mid S^M\right) = \prod_{\forall s \in S^M} p\left(l \mid s\right)$$

for the chance model independent of spike sequences, $p(l \mid S^M) = 1/dt_0$, where $dt_0$ is the sampling rate of licking events (*Figure 5—figure supplement 4B*). Finally, the total log-likelihood of the model $M$ was computed for the entire licking events $L$ as

$$\log p\left(M\right) = \sum_{\forall l \in L} \log p\left(l \mid S^M\right)$$

As a result, synchronous spikes of topTC1-2 neurons predicted occurrence of lick events (total log-likelihood for all lick events, –32541 and –7603 for TC1 in HIT trials and TC2 in FA trials, respectively) statistically better than all spikes of topTC1-2 neurons (–32825 and –7657), all spikes of all neurons in the same recording session (–33494 and –7751) and the chance level for which no correlation between spike and lick events was assumed (–35312 and –8146). Note that probabilities were trial-wise and that all lick and spike events were sampled in 0–1 second after cue onset.

## Principal component analysis

We applied principal component analysis (PCA) for a quantitative investigation of dimension reduction by the cerebellum in Go/No-go tasks. We first concatenated CS PSTHs across the four cue-response conditions for individual neurons (*Figure 6—figure supplement 1A*). Then PCA was conducted for the data matrix of 6445 neurons x 200 time-points to estimate the variance-explained-for (VAF) by the top $k$ dimensions:

$$\text{VAF}_k = \sum_{i=1}^{k} \hat{\lambda}_i$$

where $\hat{\lambda}_i = \lambda_i / \sum \lambda_i$ are fractions of the total amount of variance explained and $\lambda_i$ are the eigenvalues of the covariance matrix. Note that this computation of VAF was different from the VAF computed by TCA. We found that d=67 dimensions, defined as $d = 1/\sum \hat{\lambda}_i^2$ explained approximately 65% of variance in PSTHs of the Go/No-go data.

We applied k-means clustering (# clusters, k=4) of neurons in the PCA space and found a good agreement in neuron classification by PCA and TCA. Specifically, the selected topTC1-4 neurons (see Methods) were separately classified as k-means clusters #2, #4, #1 and #3, respectively, with a little overlap between TC2-TC4 (*Figure 6—figure supplement 1B*).

We further compared the results of PCA at the basis of single-trial and trial-averaged. In particular, we computed PSTHs for single trials (sampled in [-0.5–2] sec after cue onset, bin size of 50ms) for all neurons, which resulted in a data matrix of 216901 trials x 50 time-points. For trial-averaged data, trials of a single neuron were averaged according to the four cue-response conditions (HIT, FA, CR and MISS), resulting in a data matrix of 25780 averaged trials x 50 time-points (6445 neurons x 4 conditions = 25,780 averaged trials). PCA for the two data matrices showed comparable VAF profiles, with the dimension d=24 and d=42 for the trial-averaged and single-trial data, respectively (*Figure 6—figure supplement 1C*). This result supported our trial-averaging approach, which diminished trial-by-trial variability while retaining comparable dimensions. Note that although the recording sessions spanned the course of learning, high correlations in PSTHs of neurons in different learning stages guaranteed that identified common components for all data were reliable (*Figure 6—figure supplement 1D*).

## Statistics

All statistical analyses were performed using MATLAB software. Unless otherwise stated, data are presented as means ± std. One-way analysis of variance (ANOVA) was used to determine whether data groups of different sizes originate from the same distribution. Significance level: n.s, p>0.05; * p<0.05; ** p<0.01; *** p<0.001; **** p<0.0001.

## Code availability

The customized MATLAB code of the analyses was hosted publicly on github, accessible via https://github.com/hoang-atr/go_nogo copy archived at *Hoang, 2023*. MATLAB implementations of spike time estimation (HA_time) can be found online (https://github.com/hoang-atr/HA_time; *Hoang, 2019*).

## Acknowledgements

HH, KK and KT were supported by Grants-in-Aid for Scientific Research in Innovative Areas (17H06313) and for Transformative Research Areas (22H05160 to MM, 22H05156 to MKawato, and 22H05161 to KK). HH and KT were partially supported by JST ERATO (JPMJER1801, "Brain-AI hybrid"). HH, MKawato, and KK were partially supported by Grant Number JP21dm0307002, JP21dm0307008, and JP19dm0207080, respectively, Japan Agency for Medical Research and

Development (AMED). MKawato was partially supported by Innovative Science and Technology Initiative for Security Grant Number JP004596, Acquisition, Technology & Logistics Agency (ATLA), Japan. MKano and KK were partially supported by Grants-in-Aid for Scientific Research (JP18H04012, JP20H05915, JP21H04785 to MKano and JP22H00460 to KK) from Japan Society for the Promotion of Science (JSPS).

# Additional information

## Funding

| Funder | Grant reference number | Author |
|---|---|---|
| Japan Society for the Promotion of Science | JP17H06313 | Huu Hoang Kazuo Kitamura Keisuke Toyama |
| Japan Society for the Promotion of Science | JP22H05160 | Masanori Matsuzaki |
| Japan Society for the Promotion of Science | JP22H05156 | Mitsuo Kawato |
| Japan Society for the Promotion of Science | JP18H04012 | Masanobu Kano |
| Japan Society for the Promotion of Science | JP20H05915 | Masanobu Kano |
| Japan Society for the Promotion of Science | JP21H04785 | Masanobu Kano |
| Japan Society for the Promotion of Science | JP22H05161 | Kazuo Kitamura |
| Japan Society for the Promotion of Science | JP22H00460 | Kazuo Kitamura |
| Japan Agency for Medical Research and Development | JP21dm0307002 | Huu Hoang |
| Japan Agency for Medical Research and Development | JP21dm0307008 | Mitsuo Kawato |
| Japan Agency for Medical Research and Development | JP19dm0207080 | Kazuo Kitamura |
| Japan Science and Technology Agency | JPMJER1801 | Huu Hoang Keisuke Toyama |
| Acquisition, Technology & Logistics Agency (ATLA) | JP004596 | Mitsuo Kawato |

The funders had no role in study design, data collection and interpretation, or the decision to submit the work for publication.

## Author contributions

Huu Hoang, Software, Formal analysis, Investigation, Visualization, Methodology, Writing – original draft, Writing – review and editing; Shinichiro Tsutsumi, Resources, Data curation, Investigation, Writing – original draft, Writing – review and editing; Masanori Matsuzaki, Masanobu Kano, Conceptualization, Funding acquisition, Investigation, Writing – original draft, Writing – review and editing; Mitsuo Kawato, Conceptualization, Supervision, Funding acquisition, Investigation, Methodology, Writing – original draft, Writing – review and editing; Kazuo Kitamura, Conceptualization, Supervision, Funding acquisition, Investigation, Writing – original draft, Writing – review and editing; Keisuke Toyama, Conceptualization, Supervision, Investigation, Methodology, Writing – original draft, Writing – review and editing

## Author ORCIDs

Huu Hoang 
Masanori Matsuzaki 
Masanobu Kano 
Kazuo Kitamura 

## Ethics

All experiments were approved by the Animal Experiment Committees of the University of Tokyo (#P08-015) and the University of Yamanashi (#A27-1), and carried out in accordance with national regulations and institutional guidelines.

## Decision letter and Author response

Decision letter https://doi.org/10.7554/eLife.86340.sa1
Author response https://doi.org/10.7554/eLife.86340.sa2

---

## Additional files

### Supplementary files

• Supplementary file 1. Summary of two-photon recordings. (**a**) The number of PCs sampled in individual AldC compartments at different learning stages. (**b**) The number of trials for each cue-response condition at different learning stages.

• MDAR checklist

### Data availability

Data analysed for all the figures are included in the manuscript and source data files. The Aldoc-tdTomato mouse line is available at RIKEN Bio Resource Center (RBRC10927).

The following previously published dataset was used:

| Author(s) | Year | Dataset title | Dataset URL | Database and Identifier |
|---|---|---|---|---|
| Sakimura K | 2015 | C57BL/6N-Aldoc<tm1(tdTomato)Ksak> | https://knowledge.brc.riken.jp/resource/animal/card?brc_no=RBRC10927&__lang__=en | Riken Bioresource Research Center, RBRC10927 |

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
