## [Editor Report]

This is an important study of changes in the amplitude and synchrony of calcium responses in Purkinje cells over the course of learning, measured across a large region of the cerebellar cortex. The evidence for the conclusions is compelling, supported by sophisticated data analysis approaches. This work has the potential to inform our understanding of the functional organization of the cerebellum and longstanding hypotheses about the role of cerebellar climbing fibers in the induction of learning and in the timing of movement.

---

## [Decision Letter]

**Decision letter after peer review:**

Thank you for submitting your article "Dynamic organization of cerebellar climbing fiber response and synchrony in multiple functional modules reduces dimensions for reinforcement learning" for consideration by *eLife*. Your article has been reviewed by 2 peer reviewers, and the evaluation has been overseen by a Reviewing Editor and Michael Frank as the Senior Editor. The following individual involved in review of your submission has agreed to reveal their identity: Martijn Schonewille (Reviewer #1).

Essential revisions:

The comparison of calcium responses and synchrony of calcium responses across a large extent of the cerebellar cortex during a single behavioral task, and over the course of learning is a major strength. The study would benefit from additional analyses to support the central claims. Also the clarity of the writing and presentation of the data could both be improved substantially.

1) Additional analysis is needed to support the central claim that the complex spike responses in Crus II during the Go/No Go auditory discrimination task are "low dimensional." This additional analysis should establish the relevant standard(s) against which dimensionality is assessed -- the observed dimensionality is "low" relative to what expectation? The additional analyses should also address the following specific concerns about factors could cause an underestimation of the true response dimensionality:

– Four TCs accounting for half of the variance still leaves the potential for plenty of additional dimensionality.

– Using trial-averaged data rather than single trial responses.

– Limited task complexity (Gao, Trautmann et al. bioRxiv 2017).

– Olivary neurons, which have a long refractory period, may have a long autocorrelation time constant, which has also been identified as a factor that could limit dimensionality of neural responses (Gao, Trautmann et al. bioRxiv 2017), although this might(?) be thought of a mechanism through which the olive accomplishes dimensionality reduction.

The speculation in the text about neural dynamics within the olive implementing (vs inheriting) the decomposition of the task-related neural responses into a few TCs is not addressable without additional experiments, and should be tempered accordingly.

2) Statistical tests should be conducted to determine the statistical significance of changes in the amount of synchronization with learning, in particular, the central claims of increased synchronization in neurons contributing to TC1, which selectively respond during "hit" trials; and decreased synchrony in neurons contributing to TC2, which selectively respond during "false alarm"/FA trials.

3) Clarify what the main advance is over Tsutsumi et al. 2019 and other work regarding the correspondence of functional classification of neurons with AldoC expression and boundaries. Substantiate claims with statistical tests, e.g., comparing the similarity of the calcium response measure(s) in neurons with same vs different AldoC expression, controlling for distance between neurons (across vs within AldoC expression boundaries). Revise Figure 6 to better illustrate the claims.

4) The figures and legends should be revised to provide sufficient information for the reader to fully understand what is being shown in each plot, because some of the basics are currently lacking, as the reviewers noted with reference to many specific examples. Also a number of plots are lacking in error bars, confidence intervals, and/or statistics. In addition, the authors should make it clear that the data in the present manuscript were not just collected in the same manner as in Tsutsumi et al. 2019, as indicated in the Methods, but are actually the very same data set, if that is the case.

5) The text should be revised to improve clarity in the following areas: a) statements about what are the central new experimental findings compared to previous work of the authors and others; b) statements about what are the conclusions that are the most solidly supported by the data, and their relation to previous work; c) distinguish speculation that goes well beyond the data (there is a lot of it) versus conclusions that are strongly supported by the data. Relating the specific findings in the paper to big concepts is a strength, but blurring the line between conclusion and speculation makes it difficult for the reader to know what to take away, and some of the speculation should be substantially toned down or removed entirely. One important point where the discussion could be improved is the text describing the major longstanding hypotheses in the field about the function of the climbing fibers in response timing vs learning, and how exactly the present data inform those ideas, compared to previous work e.g., Kitazawa et al. Nature 1998.

*Reviewer #1 (Recommendations for the authors):*

Overall, this manuscript addresses an important question with an elegant technical approach and careful analysis. While the dataset is strong, there are some important points that I think could be addressed to improve the quality of the manuscript as a whole:

In the discussion, I think it would be interesting to relate these finding to those in eyeblink conditioning, for a couple of reasons. First, it would be important to discuss to what extent the go/no-go task is more 'cognitive' than eg. eyeblink conditioning. If I understand correctly, mice can start licking from cue onset, and only have to discriminate between the frequencies (if this is not the case, what happened if mice licked before the response period started in Go trials? Were these trials included, was there a timeout of 4.5 s as well?). Given the similarities, I think it would be interesting to also compare the results to those obtained using eyeblink conditioning. Eg. regarding the statement in line 433-437 suggesting that less simple spikes cause less lick, while in eyeblink conditioning fewer SSs actually drive the movement, instead of blocking the erroneous movement. Similar for lines 498-501, in which the authors explain how synchronized CSs drive a decrease in CN activity.

*Reviewer #2 (Recommendations for the authors):*

As noted above, the trial-averaged TCA makes me wonder just how similar climbing fibers from different recording sessions truly are, beyond similarities in trial-averaged activity. While a single-trial analysis is of course not possible here, I wonder if the authors can quantify the extent to which changes across learning in, for example, TC1 neurons in one zone of crus II correlate with TC1 neurons (from distinct recording sessions) in another zone of crus II, but for the same animal. This might lend additional credence to the appropriateness of trial-averaged TCA for extracting common components from non-simultaneous recording sessions.

As noted above, the idea that the several dominant tensor components reflect an olivary decomposition of task dynamics is not exactly tested in the present manuscript. The most natural test would have been to modify the behavior to clearly add or subtract a relevant task dimension, and then to examine the resulting crus II climbing fiber decomposition. However, even absent this more direct test, the authors might consider analyzing inter-subject variability. Are differences between mice in behavioral performance or behavioral strategy reflected in a different relative presence of different TCs?

As also noted, I am curious whether the authors can glean additional information about the role of increased vs decreased CF synchronization in learning. While it is likely implausible to causally test this idea, there may be interesting instrumental analyses that can better characterize the relationship of each to behavior and learning. For individual subjects, does synchronization within TC1 increase at roughly the same rate during learning that synchronization within TC2 decreases? Are the magnitudes of TC1 v TC2 synchrony changes (anti-)correlated for individual subjects?

---

## [Author Response]

Essential revisions:The comparison of calcium responses and synchrony of calcium responses across a large extent of the cerebellar cortex during a single behavioral task, and over the course of learning is a major strength. The study would benefit from additional analyses to support the central claims. Also the clarity of the writing and presentation of the data could both be improved substantially.1) Additional analysis is needed to support the central claim that the complex spike responses in Crus II during the Go/No Go auditory discrimination task are "low dimensional." This additional analysis should establish the relevant standard(s) against which dimensionality is assessed -- the observed dimensionality is "low" relative to what expectation? The additional analyses should also address the following specific concerns about factors could cause an underestimation of the true response dimensionality:– Four TCs accounting for half of the variance still leaves the potential for plenty of additional dimensionality.

We agree that we needed objective quantification of ‘low dimensionality’ of CF responses. For quantitative estimation of the dimension reduction, we shuffled the spiking data among all recorded neurons in all trials. We then constructed the same analyses of PSTHs and TCA for the shuffled data as was used for the original data (see Methods for details). The result showed that about 20 components are needed to explain approximately 50% of the variance of the shuffled data, indicating that the dimension of the real data was reduced five-fold compared with the shuffled data (Figure 3 —figure supplement 1A). We also agree that as the reviewer argued, 50% variance still leaves many potential dimensions, but selection of the four TCs was objectively determined by stability of TCA solutions (Figure 3 —figure supplement 1B). Furthermore, the remaining 50% of the variance not explained by the four TCS doesn't seem functionally important because increasing the number of components separated the components with slight differences only in temporal profiles (Figure 3 —figure supplement 1C). It is also important that the four selected TCs were functionally relevant, as shown by strong correlations of neuronal activity and behavior.

Estimation of the neural-task complex (NTC) using PCA proposed in Gao et al. (2017) is an efficient way to find upper bounds of the dimensionality of neural data, but it requires smoothly evolving neural activity patterns and continuous stimuli. For Go/No-go data, stimuli (Go and No-go cues) were discrete, resulting in four separate cue-response conditions (HIT, FA, CR, and MISS). Furthermore, neural activity patterns change within a recording session. Still, as a validation of our approach, we conducted conventional PCA for the Go/No-go data and found a good consistency between PCA and TCA. Briefly, the tensor vector 6445 x 4 x 50 was concatenated across cue-response conditions so that the input vector became 6445 x 200 (Figure 6 —figure supplement 1A). PCA of this 6445 x 200 vector showed that the Go/No-go data was low dimensional, i.e. d=67 dimensions, estimated as in Mazzucato et al. (2016), explained 65% of variance. Altogether these analyses supported our claim that Go/No-go data is low dimensional relative to the >6,000 neurons recorded. Interestingly, k-means clustering of the PC scatter further confirmed the similarity in neuron classification by PCA and TCA (Figure 6 —figure supplement 1B).

– Using trial-averaged data rather than single trial responses.

To address the reviewer’s comment regarding trial-averaged data, we conducted PCA on a trial basis and found that VAF profiles of trial-averaged and single-trial data were comparable, with 24 and 42 dimensions, respectively (Figure 6 —figure supplement 1C). This result supported our trial-averaging approach, which diminished trial-by-trial variability while retaining comparable dimensions.

– Limited task complexity (Gao, Trautmann et al. bioRxiv 2017).

From the cerebellum's viewpoint, the number of degrees-of-freedom in Go/No-go tasks is huge. For the output dimension, we could approximate the number of relevant deep cerebellar neurons as 10,000. Auditory sensory inputs to the cerebellum representing different frequencies could be at least 1,000. Relevant timing of licks with respect to auditory cues could be 100. The number of liquid rewards could vary from 0 to 5. Then, the degrees-of-freedom are 10,000 x 1,000 x 100 x 6 = 600,0000,000. This is vastly larger than the number of recorded >6000 neurons. Therefore, dimension reduction is crucial for mice to learn the Go/No-go tasks in hundreds of trials.

– Olivary neurons, which have a long refractory period, may have a long autocorrelation time constant, which has also been identified as a factor that could limit dimensionality of neural responses (Gao, Trautmann et al. bioRxiv 2017), although this might(?) be thought of a mechanism through which the olive accomplishes dimensionality reduction.

Our previous study of spontaneous IO activity showed that the increased time constant of auto-correlation would indeed increase the dimensionality (Hoang et al., PLOS Comp. Biol. 2020). More specifically, following an injection of carbenoxolone, which blocked IO gap-junctions, both the dimensionality and auto-correlation of IO neurons increased. By contrast, following an injection of picrotoxin, a GABAergic blocker, the dimensionality and auto-correlation decreased (Figure 2&3, Hoang et al., PLOS Comp. Biol. 2020). Importantly, in that study, we found that CS synchrony controls the dimensionality of IO neurons. This was also supported by the present study, showing decreased and increased dimensionality of TC1 and TC2 neurons after learning, respectively, in directions opposite to changes in their CS synchrony (Figure 3 —figure supplement 1D). We mentioned these points in the Discussion.

The speculation in the text about neural dynamics within the olive implementing (vs inheriting) the decomposition of the task-related neural responses into a few TCs is not addressable without additional experiments, and should be tempered accordingly.

We agree with the editor's comment, thus we removed Figure 7 and revised the Discussion accordingly.

2) Statistical tests should be conducted to determine the statistical significance of changes in the amount of synchronization with learning, in particular, the central claims of increased synchronization in neurons contributing to TC1, which selectively respond during "hit" trials; and decreased synchrony in neurons contributing to TC2, which selectively respond during "false alarm"/FA trials.

The present Figure 4E plots the accumulated cross-correlograms (CCGs) of selected TC1/TC2 neurons; thus, statistical tests were not applicable. To address the reviewer’s comment, we estimated synchrony strength in HIT trials (+/- 10 ms around the center bin of CCGs) of selected TC1 neurons for the three learning stages. Statistical tests confirmed a significant increase in synchrony strength during learning. Similarly, we found a significant decrease of synchrony strength in FA trials of TC2 neurons. We included histogram plots of synchrony strength for TC1/TC2 neurons in the revised Figure 4E.

3) Clarify what the main advance is over Tsutsumi et al. 2019 and other work regarding the correspondence of functional classification of neurons with AldoC expression and boundaries.

The significance of our work was calcium imaging of CF inputs spanning the entire dorsal surface of Crus II in a reward-driven Go/No-go learning task. This allowed us to study functional differences between Aldolase-C compartments in a single task that requires various cerebellar functions. Our previous study (Tsutsumi et al. 2019) suggested that distinct characteristics of CF inputs can be presented simultaneously in multiple cerebellar compartments. However, because analyses in that study were performed for calcium responses with a relatively low temporal resolution (less than 10 Hz) and after learning was completed, i.e. after mice reached expert level, it remained unclear which neural mechanisms are important and how they are organized as cerebellar components as the learning progresses. In the present study, we expanded the analysis of CF inputs throughout the course of learning. Furthermore, timing of complex spikes (CSs) was estimated at 100 Hz resolution using the hyperacuity algorithm. Then trial-averaged CS activities were decomposed using tensor component analysis. We discussed these points in the revised manuscript.

Substantiate claims with statistical tests, e.g., comparing the similarity of the calcium response measure(s) in neurons with same vs different AldoC expression, controlling for distance between neurons (across vs within AldoC expression boundaries). Revise Figure 6 to better illustrate the claims.

For quantification of differences in calcium response with respect to AldoC expression and boundaries, we compared, with statistical tests, the firing rate of lateral vs. medial and AldC positive vs. negative zones in various conditions of the cue-response condition, learning stages, and temporal windows. The results indicated that the entire CSs in Go/No-go data possesses several distinct components, each with unique firing properties. These components were determined mainly by the cue-response condition and temporal window, and moderately dependent on AldC expression and boundaries. From this viewpoint, we included these results in the revised Figures 2E-I as a motivation for the application of TCA to identify those components.

We also investigated the CS synchrony between neurons across and within AldoC expression. Synchrony strength was estimated as a function of intercellular distance and response similarity, defined by the cosine distance of the 4 TC coefficient vectors. We found significant negative correlations of CS synchrony vs. cellular/TC distances (Figures 6D-E). When grouping neurons by AldoC expression, we found a tendency for higher CS synchrony and larger response similarity for “within-zone” compared to those of “across-zone” (Figure 6F).

4) The figures and legends should be revised to provide sufficient information for the reader to fully understand what is being shown in each plot, because some of the basics are currently lacking, as the reviewers noted with reference to many specific examples. Also a number of plots are lacking in error bars, confidence intervals, and/or statistics.

We added the s.e.m and statistics particularly for Figures 1 B&D, 2K, 4D. We also improved the figures in general by providing more information in legends.

In addition, the authors should make it clear that the data in the present manuscript were not just collected in the same manner as in Tsutsumi et al. 2019, as indicated in the Methods, but are actually the very same data set, if that is the case.

Tsutsumi et al. (2019) analyzed only one-third of the Go/No-go data, i.e. after mice reached expert level, n = 79 sessions, while our paper analyzed all 6,445 neurons during the course of learning (n = 236 sessions). We stated this in the Methods, Results, and Discussion in the revised manuscript.

5) The text should be revised to improve clarity in the following areas: a) statements about what are the central new experimental findings compared to previous work of the authors and others; b) statements about what are the conclusions that are the most solidly supported by the data, and their relation to previous work; c) distinguish speculation that goes well beyond the data (there is a lot of it) versus conclusions that are strongly supported by the data. Relating the specific findings in the paper to big concepts is a strength, but blurring the line between conclusion and speculation makes it difficult for the reader to know what to take away, and some of the speculation should be substantially toned down or removed entirely.

Three main new findings were drawn based on our technical advances. First, we found that all CS activity can be decomposed into only four components (TC1-4) which capture key features of behavior and learning. This is a significant dimensional reduction performed by the cerebellum. We also found that compartmental representations of these populations align well with functional and anatomical boundaries between medial and lateral parts of Crus II, as well as expression of Aldolase-C. Second, CS synchrony, measured at the single-trial level, was concentrated among neurons that belong to the same components. Furthermore, synchrony strength in different components may change in the opposite direction during learning, suggesting that CS synchrony is the neural mechanism that drives organization of those components. Third, we demonstrated across-animal correlation, suggesting that increase of CS synchrony in TC1 neurons behaviorally decreases the lick fluctuation in Go trials. For cognitive learning, across-animal correlation was found, which suggests that a decrease of the relative presence of TC2 neurons decreases cognitive error of No-go cues across animals. Together, these results indirectly imply causal relationships of TC1 with timing control and TC2 with cognitive error learning. In summary, our study suggested that bi-directional synchronous response-associated changes in CF activities, finely constructed on compartmental structure, could reduce dimensions of the learning space, to provide a flexible learning scheme in diverse cerebellar functions. We clearly stated all these new findings at the beginning of the Discussion.

Also in the revised manuscript, we included some results to strengthen arguments about dimension reduction (Figure 3 —figure supplement 1DE, Figure 6 —figure supplement 1) and bi-directional changes in synchronization (Figure 4E). In the Discussion, as suggested by the reviewers, we summarized findings that were solidly supported by data and their related interpretations. Speculations that went beyond data were substantially toned down, e.g. transformation of TC2 to TC4, or removed entirely, e.g self-organization of components, Figure 7.

One important point where the discussion could be improved is the text describing the major longstanding hypotheses in the field about the function of the climbing fibers in response timing vs learning, and how exactly the present data inform those ideas, compared to previous work e.g., Kitazawa et al. Nature 1998.

In the revised manuscript, we expanded the discussion about possible functions of TC1 and TC2 in timing control and error-based learning. From a conceptual standpoint, these findings suggest a new direction for future investigations of climbing fiber activity. That is, populations of PCs may optimize distinct sub-goals using their own neural mechanisms, but they may collectively serve a single task.

On one hand, our analysis was consistent with previous studies that demonstrated mixed representation of error, motor, and reward parameters in CF inputs (Kitazawa et al. 1998; Markanday et al. 2021; Sendhilnathan et al. 2021; Ikezoe et al. 2022). However, unlike previous studies, multiplexing of CFs in the present study was derived solely from TC coefficients of neurons. On the other hand, our analysis further suggested a duality of motor control and cognitive learning in TC2 neurons, which has never been reported. More specifically, beside their correlation with error signals, synchronized CSs of TC2 neurons may also convey motor commands to control early licks in No-go trials. This finding implies that synchronized CSs of TC2 neurons do not simply multiplex complementary behavioral information. Rather, they contribute to generation of motor commands in downstream systems to inhibit licking following No-go cues. We discussed those points in the revised manuscript.

Reviewer #1 (Recommendations for the authors):Overall, this manuscript addresses an important question with an elegant technical approach and careful analysis. While the dataset is strong, there are some important points that I think could be addressed to improve the quality of the manuscript as a whole:In the discussion, I think it would be interesting to relate these finding to those in eyeblink conditioning, for a couple of reasons. First, it would be important to discuss to what extent the go/no-go task is more 'cognitive' than eg. eyeblink conditioning. If I understand correctly, mice can start licking from cue onset, and only have to discriminate between the frequencies (if this is not the case, what happened if mice licked before the response period started in Go trials? Were these trials included, was there a timeout of 4.5 s as well?). Given the similarities, I think it would be interesting to also compare the results to those obtained using eyeblink conditioning. Eg. regarding the statement in line 433-437 suggesting that less simple spikes cause less lick, while in eyeblink conditioning fewer SSs actually drive the movement, instead of blocking the erroneous movement. Similar for lines 498-501, in which the authors explain how synchronized CSs drive a decrease in CN activity.

Behaviorally, Go/No-go tasks are more complex than eye-blink conditioning. For the former, mice need to distinguish two types of stimuli (Go and No-go cues) by two types of response (lick or no-lick), resulting in four types of cue-response conditions (HIT, FA, CR and MISS). For eye-blink conditioning, temporal association of conditioned stimulus, e.g tone, with unconditioned stimulus, e.g. air-puff, is required to produce a timely conditioned response, e.g blink. Thus Go/No-go and eye-blink conditioning may contain both cognitive and motor elements. However, the cognitive element is more important for Go/No-go tasks, because the reward acquisition depends on correct cue discrimination and improved motor skills only maximizes the reward amount for correct responses. By contrast, in eye-blink conditioning, the movement requires more temporal precision; thus, the motor element is more important.

We agree with the reviewer that while suppression of SSs drives the movement in eye-blink conditioning (ten Brinke et al., Cell Reports, 2015), it might block erroneous movement following No-go cues. We discussed these points in the revised manuscript.

Reviewer #2 (Recommendations for the authors):As noted above, the trial-averaged TCA makes me wonder just how similar climbing fibers from different recording sessions truly are, beyond similarities in trial-averaged activity. While a single-trial analysis is of course not possible here, I wonder if the authors can quantify the extent to which changes across learning in, for example, TC1 neurons in one zone of crus II correlate with TC1 neurons (from distinct recording sessions) in another zone of crus II, but for the same animal. This might lend additional credence to the appropriateness of trial-averaged TCA for extracting common components from non-simultaneous recording sessions.

To address this comment, for individual animals, we computed the Pearson correlation of PSTHs for TC1 neurons in HIT trials and TC2 neurons in FA trials, with neurons being further divided into two pools representing “within zone” and “across zone”. Note that here TC1/TC2 neurons were sampled from all recording sessions/learning stages. The baseline level was computed from all neurons in the same recording sessions. The results indicated that correlations between TC1/TC2 neurons were significantly higher than baseline for both conditions, supporting our TCA to extract common components from non-simultaneous recording sessions. We added this result in Figure 6 —figure supplement 1D.

We further note that although TCA was conducted for trial-averaged data (PSTHs), analysis of CS synchrony was on a trial basis. The fact that CS synchrony was higher with the same TCs than across TCs (Figure 4A) indicated a strong similarity in firing activity of neurons of the same TCs on a trial basis.

As noted above, the idea that the several dominant tensor components reflect an olivary decomposition of task dynamics is not exactly tested in the present manuscript. The most natural test would have been to modify the behavior to clearly add or subtract a relevant task dimension, and then to examine the resulting crus II climbing fiber decomposition. However, even absent this more direct test, the authors might consider analyzing inter-subject variability. Are differences between mice in behavioral performance or behavioral strategy reflected in a different relative presence of different TCs?

For implementation of this idea, we first computed the fraction of TC1/TC2 neurons as functions of behavioral performance, i.e. fraction correct for Go, and fraction incorrect for No-go. As expected, we found positive correlations, i.e. fraction correct for Go increases and fraction of TC1 neurons increases, and vice versa for TC2-Nogo, for most animals. When all 17 animals combined, the increasing rate of the TC1 fraction during learning equals the decreasing rate of the TC2 fraction (Figure 5 —figure supplement 3). Strikingly, by analyzing inter-subject variability as the reviewer suggested, we found that change in a fraction of TC2 neurons was positively correlated with the change in fraction incorrect of No-go cues across animals. This result indirectly suggested a causal relationship of TC2 with cognitive error learning. We included this important result in Figure 5F and mentioned this finding in the Abstract, Results and Discussion of the revised manuscript.

As also noted, I am curious whether the authors can glean additional information about the role of increased vs decreased CF synchronization in learning. While it is likely implausible to causally test this idea, there may be interesting instrumental analyses that can better characterize the relationship of each to behavior and learning. For individual subjects, does synchronization within TC1 increase at roughly the same rate during learning that synchronization within TC2 decreases? Are the magnitudes of TC1 v TC2 synchrony changes (anti-)correlated for individual subjects?

As in the previous analysis, we computed the synchrony strength of a single neuron with other neurons in the same recording session and plotted it as a function of behavior performance. While an increase of TC1 synchronization during learning was found for most animals, the decrease of TC2 synchronization was unclear. For individual animals, the magnitudes of TC1 and TC2 synchrony changes are not correlated.

**Author response image 1. sa2fig1:** Synchrony strength of TC1 (cyan crosses) and TC2 (magenta crosses) neurons as function of fraction correct for Go cues (cyan lines) and fraction incorrect for No-go cues (magenta lines), respectively. Each panel corresponds to a single animal. The scatter at the lower-right panel showed the rate of change (i.e. the slope of synchrony strength vs. performance) in TC1 synchronization (abscissa) and the rate of change in TC2 synchronization (ordinate) of 17 animals. Each black dot represents a single animal.

Consistent with previous studies (Welsh et al. 1995; Hoogland et al. 2015; Tsutsumi et al. 2020; Wagner et al. 2021), we hypothesized that TC1 synchronization is more important for timing control than for cognitive learning. We tested this hypothesis and found that across 17 animals, the amount of change in instantaneous synchrony of TC1 neurons was significantly and negatively correlated with the amount of change in lick-latency fluctuation in single Go trials (Figure 5E). Again, this finding indirectly suggested a causal relationship that CFs projecting to TC1 neurons convey “timing signals” to control early licks, because more synchronization could stabilize the timing of motor commands by canceling noisy synaptic inputs to the IO, possibly leading to more precise timing control with less fluctuation.